# Finite-Time Analysis of Round-Robin Kullback-Leibler Upper Confidence Bounds for Optimal Adaptive Allocation with Multiple Plays and Markovian Rewards

**Vrettos Moulos**
University of California Berkeley
vrettos@berkeley.edu

## Abstract

We study an extension of the classic stochastic multi-armed bandit problem which involves multiple plays and Markovian rewards in the rested bandits setting. In order to tackle this problem we consider an adaptive allocation rule which at each stage combines the information from the sample means of all the arms, with the Kullback-Leibler upper confidence bound of a single arm which is selected in round-robin way. For rewards generated from a one-parameter exponential family of Markov chains, we provide a finite-time upper bound for the regret incurred from this adaptive allocation rule, which reveals the logarithmic dependence of the regret on the time horizon, and which is asymptotically optimal. For our analysis we devise several concentration results for Markov chains, including a maximal inequality for Markov chains, that may be of interest in their own right. As a byproduct of our analysis we also establish asymptotically optimal, finite-time guarantees for the case of multiple plays, and i.i.d. rewards drawn from a one-parameter exponential family of probability densities. Additionally, we provide simulation results that illustrate that calculating Kullback-Leibler upper confidence bounds in a round-robin way, is significantly more efficient than calculating them for every arm at each round, and that the expected regrets of those two approaches behave similarly.

## 1 Introduction

In this paper we study a generalization of the stochastic multi-armed bandit problem, where there are $K$ independent arms, and each arm $a \in [K] = \{1, \ldots, K\}$ is associated with a parameter $\theta_a \in \mathbb{R}$, and modeled as a discrete time stochastic process governed by the probability law $\mathbb{P}_{\theta_a}$. A time horizon $T$ is prescribed, and at each round $t \in [T] = \{1, \ldots, T\}$ we select $M$ arms, where $1 \leq M \leq K$, without any prior knowledge of the statistics of the underlying stochastic processes. The $M$ stochastic processes that correspond to the selected arms evolve by one time step, and we observe this evolution through a reward function, while the stochastic processes for the rest of the arms stay frozen, i.e. we consider the rested bandits setting. Our goal is to select arms in such a way so as to make the cumulative reward over the whole time horizon $T$ as large as possible. For this task we are faced with an exploitation versus exploration dilemma. At each round we need to decide whether we are going to exploit the best $M$ arms according to the information that we have gathered so far, or we are going to explore some other arms which do not seem to be so rewarding, just in case that the rewards we have observed so far deviate significantly from the expected rewards. The answer to this dilemma is usually coming by calculating indices for the arms and ranking them according to those indices, which should incorporate both information on how good an arm seems to be as well as on how many

times it has been played so far. Here we take an alternative approach where instead of calculating the indices for all the arms at each round, we just calculate the index for a single arm in a round-robin way.

## 1.1 Contributions

1. We first consider the case that the $K$ stochastic processes are irreducible Markov chains, coming from a one-parameter exponential family of Markov chains. The objective is to play as much as possible the $M$ arms with the largest stationary means, although we have no prior information about the statistics of the $K$ Markov chains. The difference of the best possible expected rewards coming from those $M$ best arms and the expected reward coming from the arms that we played is the regret that we incur. To minimize the regret we consider an index based adaptive allocation rule, Algorithm 1, which is based on sample means and Kullback-Leibler upper confidence bounds for the stationary expected rewards using the Kullback-Leibler divergence rate. We provide a finite-time analysis, Theorem 1, for this KL-UCB adaptive allocation rule which shows that the regret depends logarithmically on the time horizon $T$, and matches exactly the asymptotic lower bound, Corollary 1.

2. In order to make the finite-time guarantee possible we devise several deviation lemmata for Markov chains. An exponential martingale for Markov chains is proven, Lemma 3, which leads to a maximal inequality for Markov chains, Lemma 1. In the literature there are several approaches that use martingale techniques either to derive Hoeffding inequalities for Markov chains [15, 30], or more generally to study concentration of measure for Markov chains [24, 25, 26, 34, 27, 8, 19, 33]. Nonetheless, they're all based either on Dynkin's martingale or on Doob's martingale, combined with coupling ideas, and there is no evidence that they can lead to maximal inequalities. Moreover, a Chernoff bound for Markov chains is devised, Lemma 2, and its relation with the work of [31] is discussed in Remark 4.

3. We then consider the case that the $K$ stochastic processes are i.i.d. processes, each corresponding to a density coming from a one-parameter exponential family of densities. We establish, Theorem 2, that Algorithm 1 still enjoys the same finite-time regret guarantees, which are asymptotically optimal. The case where Theorem 2 follows directly from Theorem 1 is discussed in Remark 2. The setting of single plays is studied in [7], but with a much more computationally intense adaptive allocation rule.

4. In Section 6 we provide simulation results illustrating the fact that round-robin KL-UCB adaptive allocation rules are much more computationally efficient than KL-UCB adaptive allocation rules, and similarly round-robin UCB adaptive allocation rules are more computationally efficient than UCB adaptive allocation rules, while the expected regrets, in each family of algorithms, behave in a similar way. This brings to light round-robin schemes as an appealing practical alternative to the mainstream schemes that calculate indices for all the arms at each round.

## 1.2 Motivation

Multi-armed bandits provide a simple abstract statistical model that can be applied to study real world problems such as clinical trials, ad placement, gambling, adaptive routing, resource allocation in computer systems etc. We refer the interested reader to the survey of [6] for more context, and to the recent books of [21, 35]. The need for multiple plays can be understood in the setting of resource allocation. Scheduling jobs to a single CPU is an instance of the multi-armed bandit problem with a single play at each round, where the arms correspond to the jobs. If there are multiple CPUs we get an instance of the multi-armed bandit problem with multiple plays. The need of a richer model which allows the presence of Markovian dependence is illustrated in the context of gambling, where the arms correspond to slot-machines. It is reasonable to try to model the assertion that if a slot-machine produced a high reward the $n$-th time played, then it is very likely that it will produce a much lower reward the $(n + 1)$-th time played, simply because the casino may decide to change the reward distribution to a much stingier one if a big reward was just produced. This assertion requires, the reward distributions to depend on the previous outcome, which is precisely captured by the Markovian reward model. Moreover, we anticipate this to be an important problem attempting to bridge classical stochastic bandits, controlled Markov chains (MDPs), and non-stationary bandits.

Table 1: Summary of relevant results about regret minimization for stochastic multi-armed bandits

| work | rewards | indices | analysis |
|------|---------|---------|----------|
| [20] | i.i.d. | round-robin KL-UCB | asymptotic |
| [3] | Markovian | round-robin KL-UCB | asymptotic |
| [1] | i.i.d. | UCB | asymptotic |
| [4] | i.i.d. | UCB | finite-time |
| [37] | Markovian | UCB | finite-time |
| [12] | i.i.d. | KL-UCB | finite-time |
| this work | Markovian | round-robin KL-UCB | finite-time |

## 1.3 Related Work

The cornerstone of the multi-armed bandits literature is the pioneering work of [20], which studies the problem for the case of i.i.d. rewards and single plays. [20] introduced the change of measure argument to derive a lower bound for the problem, as well as round robin adaptive allocation rules based on upper confidence bounds which are proven to be asymptotically optimal. [2] extend the results of [20] to the case of i.i.d. rewards and multiple plays, while [1] considers index based allocation rules which are only based on sample means and are computationally simpler, although they may not be asymptotically optimal. The work of [1] inspired the first finite-time analysis for the adaptive allocation rule called UCB by [4], which is though asymptotically suboptimal. The works of [7, 12, 23] bridge this gap by providing the KL-UCB adaptive allocation rule, with finite-time guarantees which are asymptotically optimal. Additionally, [18] study a Thompson sampling algorithm for multiple plays and binary rewards, and they establish a finite-time analysis which is asymptotically optimal. Here we close the problem of multiple plays and rewards coming from an exponential family of probability densities by showing finite-time guarantees which are asymptotically optimal, via adaptive allocation rules which are much more efficiently computable than their precursors.

The study of Markovian rewards and multiple plays in the rested setting, is initiated in the work of [3]. They report an asymptotic lower bound, as well as a round robin upper confidence bound adaptive allocation rule which is proven to be asymptotically optimal. However, it is unclear if the statistics that they use in order to derive the upper confidence bounds, in their Theorem 4.1, can be recursively computed, and the practical applicability of their results is therefore questionable. In addition, they don't provide any finite-time analysis, and they use a different type of assumption on their one-parameter family of Markov chains. In particular, they assume that their one-parameter family of transition probability matrices is log-concave in the parameter, equation (4.1) in [3], while we assume that it is a one-parameter exponential family of transition probability matrices. [37, 38] extend the UCB adaptive allocation rule of [4], to the case of Markovian rewards and multiple plays. They provide a finite-time analysis, but their regret bounds are suboptimal. Moreover they impose a different type of assumption on their configuration of Markov chains. They assume that the transition probability matrices are reversible, so that they can apply Hoeffding bounds for Markov chains from [14, 22]. In a recent work [30] developed a Hoeffding bound for Markov chains, which does not assume any conditions other than irreducibility, and using this he extended the analysis of UCB to an even broader class of Markov chains. One of our main contributions is to bridge this gap and provide a KL-UCB adaptive allocation rule, with a finite-time guarantee which is asymptotically optimal. In a different line of work [32, 38] consider the restless bandits Markovian reward model, in which the state of each arm evolves according to a Markov chain independently of the player's action. Thus in the restless setting the state that we next observe is now dependent on the amount of time that elapses between two plays of the same arm.

## 2 Problem Formulation

### 2.1 One-Parameter Family of Markov Chains

We consider a one-parameter family of irreducible Markov chains on a finite state space $S$. Each member of the family is indexed by a parameter $\theta \in \mathbb{R}$, and is characterized by an initial distribution $q_\theta = [q_\theta(x)]_{x \in S}$, and an irreducible transition probability matrix $P_\theta = [P_\theta(x, y)]_{x, y \in S}$, which give rise to a probability law $\mathbb{P}_\theta$. There are $K \geq 2$ arms, with overall parameter configuration

$\boldsymbol{\theta} = (\theta_1, \ldots, \theta_K) \in \mathbb{R}^K$, and each arm $a \in [K] = \{1, \ldots, K\}$ evolves internally as the Markov chain with parameter $\theta_a$ which we denote by $\{X_n^a\}_{n \in \mathbb{Z}_{\geq 0}}$. There is a common noncostant real-valued reward function on the state space $f : S \to \mathbb{R}$, and successive plays of arm $a$ result in observing samples from the stochastic process $\{Y_n^a\}_{n \in \mathbb{Z}_{\geq 0}}$, where $Y_n^a = f(X_n^a)$. In other words, the distribution of the rewards coming from arm $a$ is a function of the Markov chain with parameter $\theta_a$, and thus it can have more complicated dependencies. As a special case, if we pick the reward function $f$ to be injective, then the distribution of the rewards is Markovian.

For $\theta \in \mathbb{R}$, due to irreducibility, there exists a unique stationary distribution for the transition probability matrix $P_\theta$ which we denote with $\pi_\theta = [\pi_\theta(x)]_{x \in S}$. Furthermore, let $\mu(\theta) = \sum_{x \in S} f(x)\pi_\theta(x)$ be the stationary mean reward corresponding to the Markov chain parametrized by $\theta$. Without loss of generality we may assume that the $K$ arms are ordered so that,

$$\mu(\theta_1) \geq \ldots \geq \mu(\theta_N) > \mu(\theta_{N+1}) \ldots = \mu(\theta_M) = \ldots = \mu(\theta_L) > \mu(\theta_{L+1}) \geq \ldots \geq \mu(\theta_K),$$

for some $N \in \{0, \ldots, M-1\}$ and $L \in \{M, \ldots, K\}$, where $N = 0$ means that $\mu(\theta_1) = \ldots = \mu(\theta_M)$, $L = K$ means that $\mu(\theta_M) = \ldots = \mu(\theta_K)$, and we set $\mu(\theta_0) = \infty$ and $\mu(\theta_{K+1}) = -\infty$.

## 2.2 Regret Minimization

We fix a time horizon $T$, and at each round $t \in [T] = \{1, \ldots, T\}$ we play a set $\phi_t$ of $M$ distinct arms, where $1 \leq M \leq K$ is the same through out the rounds, and we observe rewards $\{Z_t^a\}_{a \in [K]}$ given by, $Z_t^a = Y_{N_a(t)}^a \cdot I\{a \in \phi_t\}$, where $N^a(t) = \sum_{s=1}^t I\{a \in \phi_s\}$ is the number of times we played arm $a$ up to time $t$. Using the stopping times $\tau_n^a = \inf\{t \geq 1 : N^a(t) = n\}$, we can also reconstruct the $\{Y_n^a\}_{n \in \mathbb{Z}_{>0}}$ process, from the observed $\{Z_t^a\}_{t \in \mathbb{Z}_{>0}}$ process, via the identity $Y_n^a = Z_{\tau_n^a}^a$. Our play $\phi_t$ is based on the information that we have accumulated so far. In other words, the event $\{\phi_t = A\}$, for $A \subseteq [K]$ with $|A| = M$, belongs to the $\sigma$-field generated by $\phi_1, \{Z_1^a\}_{a \in [K]}, \ldots, \phi_{t-1}, \{Z_{t-1}^a\}_{a \in [K]}$. We call the sequence $\boldsymbol{\phi} = \{\phi_t\}_{t \in \mathbb{Z}_{>0}}$ of our plays an *adaptive allocation rule*. Our goal is to come up with an adaptive allocation rule $\boldsymbol{\phi}$, that achieves the greatest possible expected value for the sum of the rewards,

$$S_T = \sum_{t=1}^T \sum_{a \in [K]} Z_t^a = \sum_{a \in [K]} \sum_{n=1}^{N^a(T)} Y_n^a,$$

which is equivalent to minimizing the expected regret,

$$R_{\boldsymbol{\theta}}^{\boldsymbol{\phi}}(T) = T \sum_{a=1}^M \mu(\theta_a) - \mathbb{E}_{\boldsymbol{\theta}}^{\boldsymbol{\phi}}[S_T]. \tag{1}$$

## 2.3 Asymptotic Lower Bound

A quantity that naturally arises in the study of regret minimization for Markovian bandits is the *Kullback-Leibler divergence rate* between two Markov chains, which is a generalization of the usual Kullback-Leibler divergence between two probability distributions. We denote by $D(\theta \parallel \lambda)$ the Kullback-Leibler divergence rate between the Markov chain with parameter $\theta$ and the Markov chain with parameter $\lambda$, which is given by,

$$D(\theta \parallel \lambda) = \sum_{x,y \in S} \log \frac{P_\theta(x,y)}{P_\lambda(x,y)} \pi_\theta(x) P_\theta(x,y), \tag{2}$$

where we use the standard notational conventions $\log 0 = \infty$, $\log \frac{\alpha}{0} = \infty$ if $\alpha > 0$, and $0 \log 0 = 0 \log \frac{0}{0} = 0$. Indeed note that, if $P_\theta(x, \cdot) = p_\theta(\cdot)$ and $P_\lambda(x, \cdot) = p_\lambda(\cdot)$, for all $x \in S$, i.e. in the special case that the Markov chains correspond to IID processes, then the Kullback-Leibler divergence rate $D(\theta \parallel \lambda)$ is equal to the Kullback-Leibler divergence $D(p_\theta \parallel p_\lambda)$ between $p_\theta$ and $p_\lambda$,

$$D(\theta \parallel \lambda) = \sum_{x,y \in S} \log \frac{p_\theta(y)}{p_\lambda(y)} p_\theta(x) p_\theta(y) = \sum_{y \in S} \log \frac{p_\theta(y)}{p_\theta(y)} p_\theta(y) = D(p_\theta \parallel p_\lambda).$$

Under some regularity assumptions on the one-parameter family of Markov chains, [3] in their Theorem 3.1 are able to establish the following asymptotic lower bound on the expected regret for

any adaptive allocation rule $\phi$ which is uniformly good across all parameter configurations,

$$\liminf_{T \to \infty} \frac{R_{\boldsymbol{\theta}}^{\boldsymbol{\phi}}(T)}{\log T} \geq \sum_{b=L+1}^{K} \frac{\mu(\theta_M) - \mu(\theta_b)}{D\left(\theta_b \parallel \theta_M\right)}. \tag{3}$$

A further discussion of this lower bound, as well as an alternative derivation can be found in Appendix D,

## 2.4 One-Parameter Exponential Family Of Markov Chains

Let $S$ be a finite state space, $f : S \to \mathbb{R}$ be a nonconstant reward function on the state space, and $P$ an irreducible transition probability matrix on $S$, with associated stationary distribution $\pi$. $P$ will serve as the generator stochastic matrix of the family. Let $\mu(0) = \sum_{x \in S} f(x)\pi(x)$ be the stationary mean of the Markov chain induced by $P$ when $f$ is applied. By tilting exponentially the transitions of $P$ we are able to construct new transition matrices that realize a whole range of stationary means around $\mu(0)$ and form the exponential family of stochastic matrices. Let $\theta \in \mathbb{R}$, and consider the matrix $\tilde{P}_\theta(x, y) = P(x, y)e^{\theta f(y)}$. Denote by $\rho(\theta)$ its spectral radius. According to the Perron-Frobenius theory, see Theorem 8.4.4 in the book of [16], $\rho(\theta)$ is a simple eigenvalue of $\tilde{P}_\theta$, called the Perron-Frobenius eigenvalue, and we can associate to it unique left $u_\theta$ and right $v_\theta$ eigenvectors such that they are both positive, $\sum_{x \in S} u_\theta(x) = 1$ and $\sum_{x \in S} u_\theta(x)v_\theta(x) = 1$. Using them we define the member of the exponential family which corresponds to the natural parameter $\theta$ as,

$$P_\theta(x, y) = \frac{v_\theta(y)}{v_\theta(x)} \exp\left\{\theta f(y) - \Lambda(\theta)\right\} P(x, y), \tag{4}$$

where $\Lambda(\theta) = \log \rho(\theta)$ is the log-Perron-Frobenius eigenvalue. It can be easily seen that $P_\theta(x, y)$ is indeed a stochastic matrix, and its stationary distribution is given by $\pi_\theta(x) = u_\theta(x)v_\theta(x)$. The initial distribution $q_\theta$ associated to the parameter $\theta$, can be any distribution on $S$, since the KL-UCB adaptive allocation rule that we devise, and its guarantees, will be valid no matter the initial distributions.

*Example* 1 (Two-state chain). Let $S = \{0, 1\}$, and consider the transition probability matrix, $P$, representing two coin-flips, $\text{Bernoulli}(p)$ when we're in state 0, and $\text{Bernoulli}(q)$ when we're in state 1. Let $f(x) = 2x - 1$ be the state reward function, and denote let $\{P_\theta : \theta \in \mathbb{R}\}$ be the exponential family of transition probability matrices generated by $P$ and $f$. Then,

$$P_0 = P = \begin{bmatrix} 1 - p & p \\ 1 - q & q \end{bmatrix}, \text{ and } P_\theta = \frac{1}{\rho(\theta)} \begin{bmatrix} (1-p)e^{-\theta} & \rho(\theta) - (1-p)e^{-\theta} \\ \rho(\theta) - qe^\theta & qe^\theta \end{bmatrix},$$

where the Perron-Frobenius eigenvalue is given by,

$$\rho(\theta) = \frac{(1-p)e^{-\theta} + qe^\theta + \sqrt{((1-p)e^{-\theta} - qe^\theta)^2 + 4p(1-q)}}{2}.$$

In the special case that $p = q$, we get back the typical exponential family of $\text{Bernoulli}(p_\theta)$ coin-flips, with

$$1 - p_\theta = \frac{(1-p)e^{-\theta}}{(1-p)e^{-\theta} + pe^\theta}.$$

Exponential families of Markov chains date back to the work of [28]. For a short overview of one-parameter exponential families of Markov chains, as well as proofs of the following properties, we refer the reader to Section 2 in [31]. The log-Perron-Frobenius eigenvalue $\Lambda(\theta)$ is a convex analytic function on the real numbers, and through its derivative, $\dot{\Lambda}(\theta)$, we obtain the stationary mean $\mu(\theta)$ of the Markov chain with transition matrix $P_\theta$ when $f$ is applied, i.e. $\dot{\Lambda}(\theta) = \mu(\theta) = \sum_{x \in S} f(x)\pi_\theta(x)$. When $\Lambda(\theta)$ is not the linear function $\theta \mapsto \mu(0)\theta$, the log-Perron-Frobenius eigenvalue, $\Lambda(\theta)$, is strictly convex and thus its derivative $\dot{\Lambda}(\theta)$ is strictly increasing, and it forms a bijection between the natural parameter space, $\mathbb{R}$, and the mean parameter space, $\mathcal{M} = \dot{\Lambda}(\mathbb{R})$, which is a bounded open interval.

The Kullback-Leibler divergence rate from (2), when instantiated for the exponential family of Markov chains, can be expressed as,

$$D\left(\theta \parallel \lambda\right) = \Lambda(\lambda) - \Lambda(\theta) - \dot{\Lambda}(\theta)(\lambda - \theta),$$

which is convex and differentiable over $\mathbb{R} \times \mathbb{R}$. Since $\dot{\Lambda} : \mathbb{R} \to \mathcal{M}$ forms a bijection from the natural parameter space, $\mathbb{R}$, to the mean parameter space, $\mathcal{M}$, with some abuse of notation we will write $D\left(\mu \parallel \nu\right)$ for $D\left(\dot{\Lambda}^{-1}(\mu) \parallel \dot{\Lambda}^{-1}(\nu)\right)$, where $\mu, \nu \in \mathcal{M}$. Furthermore, $D\left(\cdot \parallel \cdot\right) : \mathcal{M} \times \mathcal{M} \to \mathbb{R}_{\geq 0}$ can be extended continuously, to a function $D\left(\cdot \parallel \cdot\right) : \bar{\mathcal{M}} \times \bar{\mathcal{M}} \to \mathbb{R}_{\geq 0} \cup \{\infty\}$, where $\bar{\mathcal{M}}$ denotes the closure of $\mathcal{M}$. This can even further be extended to a convex function on $\mathbb{R} \times \mathbb{R}$, by setting $D\left(\mu \parallel \nu\right) = \infty$ if $\mu \notin \bar{\mathcal{M}}$ or $\nu \notin \bar{\mathcal{M}}$. For fixed $\nu \in \mathbb{R}$, the function $\mu \mapsto D\left(\mu \parallel \nu\right)$ is decreasing for $\mu \leq \nu$ and increasing for $\mu \geq \nu$. Similarly, for fixed $\mu \in \mathbb{R}$, the function $\nu \mapsto D\left(\mu \parallel \nu\right)$ is decreasing for $\nu \leq \mu$ and increasing for $\nu \geq \mu$.

# 3 A Maximal Inequality for Markov Chains

The following definition is the technical condition that we will require for our maximal inequality.

*Definition* 1 (Doeblin's type of condition). Let $P$ be a transition probability matrix on the finite state space $S$. For a nonempty set of states $A \subset S$, we say that $P$ is $A$-Doeblin if, the submatrix of $P$ with rows and columns in $A$ is irreducible, and for every $x \in S - A$ there exists $y \in A$ such that $P(x, y) > 0$.

*Example* 1 (continued). For this example $P$ being $\{0\}$-Doeblin means that $p, q \in [0, 1)$, but already irreducibility requires that $p \in (0, 1]$ and $q \in [0, 1)$, hence the only additional constraint is $p \neq 1$.

*Remark* 1. Our Definition 1 is inspired by the classic Doeblin's Theorem, see Theorem 2.2.1 in [36]. Doeblin's Theorem states that, if the transition probability matrix $P$ satisfies Doeblin's condition (namely there exists $\epsilon > 0$, and a state $y \in S$ such that for all $x \in S$ we have $P(x, y) \geq \epsilon$), then $P$ has a unique stationary distribution $\pi$, and for all initial distributions $q$ we have geometric convergence to stationarity, i.e. $\|qP^n - \pi\|_1 \leq 2(1 - \epsilon)^n$. Doeblin's condition, according to our Definition 1, corresponds to $P$ being $\{y\}$-Doeblin for some $y \in S$.

**Lemma 1** (Maximal inequality for irreducible Markov chains satisfying Doeblin's condition). *Let $\{X_n\}_{n \in \mathbb{Z}_{\geq 0}}$ be an irreducible Markov chain over the finite state space $S$ with transition matrix $P$, initial distribution $q$, and stationary distribution $\pi$. Let $f : S \to \mathbb{R}$ be a non-constant function on the state space. Denote by $\mu(0) = \sum_{x \in S} f(x)\pi(x)$ the stationary mean when $f$ is applied, and by $\bar{Y}_n = \frac{1}{n} \sum_{k=1}^{n} Y_k$ the empirical mean, where $Y_k = f(X_k)$. Assume that $P$ is $(\arg\min_{x \in S} f(x))$-Doeblin. Then for all $\epsilon > 1$ we have*

$$\mathbb{P}\left( \bigcup_{k=1}^{n} \left\{ \mu(0) \geq \bar{Y}_k \text{ and } kD\left(\bar{Y}_k \parallel \mu(0)\right) \geq \epsilon \right\} \right) \leq C_- e \lceil \epsilon \log n \rceil e^{-\epsilon},$$

*where $C_- = C_-(P, f)$ is a positive constant depending only on the transition probability matrix $P$ and the function $f$.*

Proof of this and other concentration lemmata can be found in Appendix A. I.i.d. versions of this maximal inequality have found applicability not only in multi-armed bandit problems, but also in the case of context tree estimation, [13], indicating that our Lemma 1 may be of interest for other applications as well.

# 4 The Round-Robin KL-UCB Adaptive Allocation Rule for Multiple Plays and Markovian Rewards

For each arm $a \in [K]$ we define the empirical mean at the global time $t$ as,

$$\bar{Y}_a(t) = (Y_1^a + \ldots + Y_{N_a(t)}^a)/N_a(t), \tag{5}$$

and its local time counterpart as,

$$\bar{Y}_n^a = (Y_1^a + \ldots + Y_n^a)/n,$$

with their link being $\bar{Y}_n^a = \bar{Y}_a(\tau_n^a)$, where $\tau_n^a = \inf\{t \geq 1 : N_a(t) = n\}$. At each round $t$ we calculate a single upper confidence bound index,

$$U_a(t) = \sup\left\{ \mu \in \mathcal{M} : D\left(\bar{Y}_a(t) \parallel \mu\right) \leq \frac{g(t)}{N_a(t)} \right\}, \tag{6}$$

where $g(t)$ is an increasing function, and we denote its local time version by,

$$U_n^a(t) = \sup\left\{\mu \in \mathcal{M} : D\left(\bar{Y}_n^a \,\|\, \mu\right) \leq \frac{g(t)}{n}\right\}.$$

Note that $U_a(t)$ is efficiently computable via a bisection method due to the monotonicity of $D\left(\bar{Y}_a(t) \,\|\, \cdot\right)$. It is straightforward to check, using the definition of $U_n^a(t)$, the following two relations,

$$\bar{Y}_n^a \leq U_n^a(t) \text{ for all } n \leq t, \tag{7}$$
$$U_n^a(t) \text{ is increasing in } t \geq n \text{ for fixed } n. \tag{8}$$

Furthermore, in Appendix B we study the concentration properties of those upper confidence indices and of the sample means, using the concentration results for Markov chains from Section 3. The idea of calculating indices in a round robin way, dates back to the seminal work of [20]. Here we exploit this idea, which seems to have been forgotten over time in favor of algorithms that calculate indices for all the arms at each time, and we augment it with the usage of the upper confidence bounds in (6), which are efficiently computable, see Section 6 for simulation results, as opposed to the statistics in Theorem 4.1 from [3]. Moreover, this combination of a round-robin scheme and the indices in (6) is amenable to a finite-time analysis, see Appendix C.

---

**Algorithm 1:** The round-robin KL-UCB adaptive allocation rule.

---

**Parameters:** *number of arms $K \geq 2$, time horizon $T \geq K$, number of plays $1 \leq M \leq K$, KL divergence rate function $D\left(\cdot \,\|\, \cdot\right) : \bar{\mathcal{M}} \times \bar{\mathcal{M}} \to \mathbb{R}_{\geq 0}$, increasing function $g : \mathbb{Z}_{>0} \to \mathbb{R}$, parameter $\delta \in (0, 1/K)$;*
**Initializaton:** *In the first $K$ rounds pull each arm $M$ times and set*
$\bar{Y}_a(K) = (Y_1^a + \ldots + Y_M^a)/M$, *for $a = 1, \ldots, K$;*
**for** $t = K, \ldots, T-1$ **do**
    Let $W_t = \{a \in [K] : N_a(t) \geq \lceil \delta t \rceil\}$;
    Pick any subset of arms $L_t \subseteq W_t$ such that:

$$|L_t| = M, \text{ and } \min_{a \in L_t} \bar{Y}_a(t) \geq \sup_{b \in W_t - L_t} \bar{Y}_b(t);$$

    Let $b \equiv t+1 \pmod{K}$, with $b \in [K]$;
    Let $U_b(t) = \sup\left\{\mu \in \mathcal{M} : D\left(\bar{Y}_b(t) \,\|\, \mu\right) \leq \dfrac{g(t)}{N_b(t)}\right\}$;
    **if** $b \in L_t$ **or** $\min_{a \in L_t} \bar{Y}_a(t) \geq U_b(t)$ **then**
        | Pull the $M$ arms in $\phi_{t+1} = L_t$;
    **else**
        Pick any $a \in \arg\min_{a \in L_t} \bar{Y}_a(t)$;
        Pull the $M$ arms in $\phi_{t+1} = (L_t \setminus \{a\}) \cup \{b\}$;
    **end**
**end**

---

**Theorem 1** (Markovian rewards and multiple plays: finite-time guarantees). *Let $P$ be an irreducible transition probability matrix on the finite state space $S$, and $f : S \to \mathbb{R}$ be a real-valued reward function, such that $P$ is $(\arg\min_{x \in S} f(x))$-Doeblin. Assume that the $K$ arms correspond to the parameter configuration $\boldsymbol{\theta} \in \mathbb{R}^K$ of the exponential family of Markov chains, as described in Equation 4. Without loss of generality assume that the $K$ arms are ordered so that,*

$$\mu(\theta_1) \geq \ldots \geq \mu(\theta_N) > \mu(\theta_{N+1}) \ldots = \mu(\theta_M) = \ldots = \mu(\theta_L) > \mu(\theta_{L+1}) \geq \ldots \geq \mu(\theta_K).$$

*Fix $\epsilon \in (0, \min(\mu(\theta_N) - \mu(\theta_M), \mu(\theta_M) - \mu(\theta_{L+1})))$. The KL-UCB adaptive allocation rule for Markovian rewards and multiple plays, Algorithm 1, with the choice $g(t) = \log t + 3\log\log t$, enjoys the following finite-time upper bound on the regret,*

$$R_{\boldsymbol{\theta}}^{\boldsymbol{\phi}}(T) \leq \sum_{b=L+1}^{K} \frac{\mu(\theta_M) - \mu(\theta_b)}{D\left(\mu(\theta_b) \,\|\, \mu(\theta_M) - \epsilon\right)} \log T + c_1\sqrt{\log T} + c_2\log\log T + c_3\sqrt{\log\log T} + c_4,$$

*where $c_1, c_2, c_3, c_4$ are constants with respect to $T$, which are given more explicitly in the analysis.*

**Corollary 1** (Asymptotic optimality). *In the context of Theorem 1 the KL-UCB adaptive allocation rule, Algorithm 1, is asymptotically optimal, and,*

$$\lim_{T \to \infty} \frac{R_{\boldsymbol{\theta}}^{\boldsymbol{\phi}}(T)}{\log T} = \sum_{b=L+1}^{K} \frac{\mu(\theta_M) - \mu(\theta_b)}{D\left(\mu(\theta_b) \parallel \mu(\theta_M)\right)}.$$

## 5  The Round-Robin KL-UCB Adaptive Allocation Rule for Multiple Plays and i.i.d. Rewards

As a byproduct of our work in Section 4 we further obtain a finite-time regret bound, which is asymptotically optimal, for the case of multiple plays and i.i.d. rewards, from an exponential family of probability densities.

We first review the notion of an exponential family of probability densities, for which the standard reference is [5]. Let $(X, \mathcal{X}, \rho)$ be a probability space. A one-parameter exponential family is a family of probability densities $\{p_\theta : \theta \in \Theta\}$ with respect to the measure $\rho$ on $X$, of the form,

$$p_\theta(x) = \exp\{\theta f(x) - \Lambda(\theta)\}h(x), \tag{9}$$

where $f : X \to \mathbb{R}$ is called the sufficient statistic, is $\mathcal{X}$-measurable, and there is no $c \in \mathbb{R}$ such that $f(x) \overset{\rho-a.s.}{=} c$, $h : X \to \mathbb{R}_{\geq 0}$ is called the carrier density, and is a density with respect to $\rho$, and $\Lambda$ is called the log-Moment-Generating-Function and is given by $\Lambda(\theta) = \log \int_X e^{\theta f(x)} h(x) \rho(dx)$, which is finite for $\theta$ in the natural parameter space $\Theta = \{\theta \in \mathbb{R} : \int_X e^{\theta f(x)} h(x) \rho(dx) < \infty\}$. The log-MGF, $\Lambda(\theta)$, is strictly convex and its derivative forms a bijection between the natural parameters, $\theta$, and the mean parameters, $\mu(\theta) = \int_X f(x) p_\theta(x) \rho(dx)$. The Kullback-Leibler divergence between $p_\theta$ and $p_\lambda$, for $\theta, \lambda \in \Theta$, can be written as $D\left(\theta \parallel \lambda\right) = \Lambda(\lambda) - \Lambda(\theta) - \dot{\Lambda}(\theta)(\lambda - \theta)$.

For this section, each arm $a \in [K]$ with parameter $\theta_a$ corresponds to the i.i.d. process $\{X_n^a\}_{n \in \mathbb{Z}_{>0}}$, where each $X_n^a$ has density $p_{\theta_a}$ with respect to $\rho$, which gives rise to the i.i.d. reward process $\{Y_n^a\}_{n \in \mathbb{Z}_{>0}}$, with $Y_n^a = f(X_n^a)$.

*Remark* 2. When there is a finite set $S \in \mathcal{X}$ such that $\rho(S) = 1$, then the exponential family of probability densities in Equation 9, is just a special case of the exponential family of Markov chains in Equation 4, as can be seen by setting $P(x, \cdot) = h(\cdot)$, for all $x \in S$. Then $v_\theta(x) = 1$ for all $x \in S$, the log-Perron-Frobenius eigenvalue coincides with the log-MGF, and $\Theta = \mathbb{R}$. Therefore, Theorem 1 already resolves the case of multiple plays and i.i.d. rewards from an exponential family of finitely supported densities.

**Theorem 2** (i.i.d. rewards and multiple plays: finite-time guarantees). *Let $(X, \mathcal{X}, \rho)$ be a probability space, $f : X \to \mathbb{R}$ a $\mathcal{X}$-measurable function, and $h : X \to \mathbb{R}_{\geq 0}$ a density with respect to $\rho$. Assume that the $K$ arms correspond to the parameter configuration $\boldsymbol{\theta} \in \Theta^K$ of the exponential family of probability densities, as described in Equation 9. Without loss of generality assume that the $K$ arms are ordered so that,*

$$\mu(\theta_1) \geq \ldots \geq \mu(\theta_N) > \mu(\theta_{N+1}) \ldots = \mu(\theta_M) = \ldots = \mu(\theta_L) > \mu(\theta_{L+1}) \geq \ldots \geq \mu(\theta_K).$$

*Fix $\epsilon \in (0, \min(\mu(\theta_N) - \mu(\theta_M), \mu(\theta_M) - \mu(\theta_{L+1})))$. The KL-UCB adaptive allocation rule for i.i.d. rewards and multiple plays, Algorithm 1, with the choice $g(t) = \log t + 3 \log \log t$, enjoys the following finite-time upper bound on the regret,*

$$R_{\boldsymbol{\theta}}^{\boldsymbol{\phi}}(T) \leq \sum_{b=L+1}^{K} \frac{\mu(\theta_M) - \mu(\theta_b)}{D\left(\mu(\theta_b) \parallel \mu(\theta_M) - \epsilon\right)} \log T + c_1 \sqrt{\log T} + c_2 \log \log T + c_3 \sqrt{\log \log T} + c_4,$$

*where $c_1, c_2, c_3, c_4$ are constants with respect to $T$.*

*Remark* 3. For the special case of single plays, $M = 1$, such a finite-time regret bound is derived in [7], and here we generalize it for multiple plays, $1 \leq M \leq K$. One striking difference is that we consider calculations of KL upper confidence bounds in a round-robin way, as opposed to calculating them for all the arms at each round. But computing KL-UCB indices adds an extra computational overhead, as it entails inverting an increasing function via the bisection method. Thus, our approach has important practical implications as it leads to significantly more efficient algorithms. In particular, if we run the bisection method until we reach accuracy $\delta$, then the KL-UCB of [7] has a cost of $O(K \log 1/\delta)$ per round, while the round-robin KL-UCB described in Algorithm 1 has a cost of $O(K + \log 1/\delta)$ per round.

# 6 Simulation Results

In the context of Example 1, we set $p = 0.49, q = 0.45, K = 14$, and $T = 10^6$. We generated the bandit instance $\theta_1, \ldots, \theta_K$ by drawing i.i.d. $N(0, 1/16)$ samples. Four adaptive allocation rules were taken into consideration:

1. **UCB**: at reach round calculate all UCB indices,

$$U_a^{\text{UCB}}(t) = \bar{Y}_a(t) + \beta\sqrt{\frac{2\log t}{N_a(t)}}, \text{ for } a = 1, \ldots, K.$$

2. **Round-Robin UCB**: at reach round calculate a single UCB index,

$$U_b^{\text{UCB}}(t) = \bar{Y}_b(t) + \beta\sqrt{\frac{2\log t}{N_b(t)}}, \text{ only for } b \equiv t+1 \pmod{K}.$$

3. **KL-UCB**: at reach round calculate all KL-UCB indices,

$$U_a^{\text{KL}-\text{UCB}}(t) = \sup\left\{\mu \in \mathcal{M} : D\left(\bar{Y}_a(t) \,\|\, \mu\right) \le \frac{\log t + 3\log\log t}{N_a(t)}\right\}, \text{ for } a = 1, \ldots, K.$$

4. **Round-Robin KL-UCB**: at reach round calculate a single KL-UCB index,

$$U_b^{\text{KL}-\text{UCB}}(t) = \sup\left\{\mu \in \mathcal{M} : D\left(\bar{Y}_b(t) \,\|\, \mu\right) \le \frac{\log t + 3\log\log t}{N_b(t)}\right\}, \text{ only for } b \equiv t+1 \pmod{K}.$$

For the UCB indices, after some tuning, we picked $\beta = 1$ which is significantly smaller than the theoretical values of $\beta$ from [37, 38, 30]. For each of those adaptive allocation rules $10^4$ Monte Carlo iterations were performed in order to estimate the expected regret, and the simulation results are presented in the following plots.

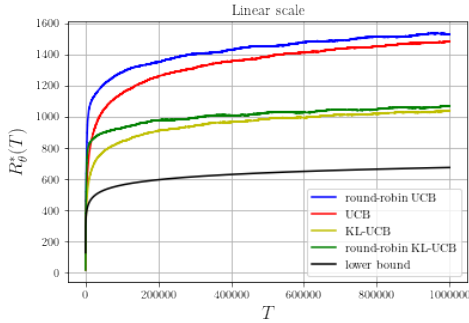

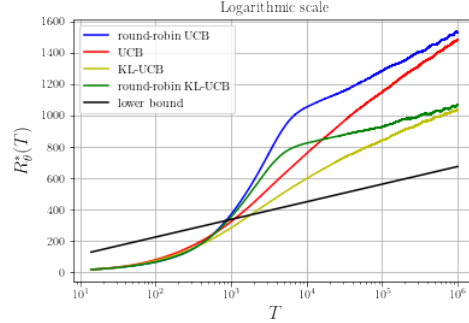

Figure 1: Regret of the various algorithms as a function of time in linear scale.

Figure 2: Regret of the various algorithms as a function of time in logarithmic scale.

For our simulations we used the programming language C, to produce highly efficient code, and a personal computer with a 2.6GHz processor and 16GB of memory. We report that the simulation for the Round-Robin KL-UCB adaptive allocation rule was $14.48$ times faster than the simulation for the KL-UCB adaptive allocation rule. This behavior is expected since each calculation of a KL-UCB index induces a significant computation cost as it involves finding the inverse of an increasing function using the bisection method. Additionally, the simulation for the Round-Robin UCB adaptive allocation rule was $3.15$ times faster than the simulation for the KL-UCB adaptive allocation rule, and this is justified from the fact that calculating mathematical functions such as $\log(\cdot)$ and $\sqrt{\cdot}$, is more costly than calculating averages which only involve a division. Our simulation results yield that in practice round-robin schemes are significantly faster than schemes that calculate the indices of all the arms at each round, and the computational gap is increasing with the number of arms $K$, while the behavior of the expected regrets is very similar.

## Statement of Broader Impact

This work touches upon a very old problem dating back to 1933 and the work of [39]. Therefore, we don't anticipate any new societal impacts or ethical aspects, that are not well understood by now.

## Acknowledgements

This research was supported in part by by the NSF grant CCF-1816861.

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
