[Supplementary Material]

## Appendix A    Concentration Lemmata for Markov Chains

We first develop a Chernoff bound, which remarkably does not impose any conditions on the Markov chain other than irreducibility, which is though a mandatory requirement for the stationary mean to be well-defined.

**Lemma 2** (Chernoff bound for irreducible Markov chains). *Let $\{X_n\}_{n\in\mathbb{Z}_{\geq 0}}$ be an irreducible Markov chain over the finite state space $S$ with transition probability matrix $P$, initial distribution $q$, and stationary distribution $\pi$. Let $f : S \to \mathbb{R}$ be a nonconstant function on the state space. Denote by $\mu(0) = \sum_{x\in S} f(x)\pi(x)$ the stationary mean when $f$ is applied, and by $\bar{Y}_n = \frac{1}{n}\sum_{k=1}^{n} Y_k$ the empirical mean, where $Y_k = f(X_k)$. Let $F$ be a closed subset of $\mathcal{M} \cap [\mu(0), \infty)$. Then,*

$$\mathbb{P}\left(\bar{Y}_n \geq \mu\right) \leq C_+ e^{-nD(\mu \,\|\, \mu(0))}, \text{ for } \mu \in F,$$

*where $D\left(\cdot \,\|\, \cdot\right)$ stands for the Kullback-Leibler divergence rate in the exponential family of stochastic matrices generated by $P$ and $f$, and $C_+ = C_+(P, f, F)$ is a positive constant depending only on the transition probability matrix $P$, the function $f$ and the closed set $F$.*

*Proof of Lemma 2.*
Using the standard exponential transform followed by Markov's inequality we obtain that for any $\theta \geq 0$,

$$\mathbb{P}(\bar{Y}_n \geq \mu) \leq \mathbb{P}(e^{n\theta\bar{Y}_n} \geq e^{n\theta\mu}) \leq \exp\left\{-n\left(\theta\mu - \frac{1}{n}\log\mathbb{E}\left[e^{\theta(f(X_1)+\ldots+f(X_n))}\right]\right)\right\}.$$

We can upper bound the expectation from above in the following way,

$$\begin{aligned}
\mathbb{E}\left[e^{\theta(f(X_1)+\ldots+f(X_n))}\right] &= \sum_{x_0,\ldots,x_n\in S} q(x_0)P(x_0,x_1)e^{\theta f(x_1)}\ldots P(x_{n-1},x_n)e^{\theta f(x_n)} \\
&= \sum_{x_0,x_n\in S} q(x_0)\tilde{P}_\theta^n(x_0,x_n) \\
&\leq \frac{1}{\min_{x\in S} v_\theta(x)} \sum_{x_0,x_n\in S} q(x_0)\tilde{P}_\theta^n(x_0,x_n)v_\theta(x_n) \\
&= \frac{\rho(\theta)^n}{\min_{x\in S} v_\theta(x)} \sum_{x_0\in S} q(x_0)v_\theta(x_0) \\
&\leq \max_{x,y\in S} \frac{v_\theta(y)}{v_\theta(x)}\rho(\theta)^n,
\end{aligned}$$

where in the last equality we used the fact that $v_\theta$ is a right Perron-Frobenius eigenvector of $\tilde{P}_\theta$.

From those two we obtain,

$$\mathbb{P}(\bar{Y}_n \geq \mu) \leq \max_{x,y\in S} \frac{v_\theta(y)}{v_\theta(x)} \exp\left\{-n(\theta\mu - \Lambda(\theta))\right\},$$

and if we plug in $\theta_\mu = \dot{\Lambda}^{-1}(\mu)$, which is a nonnegative real number since $\mu \in F \subseteq \mathcal{M} \cap [\mu(0), \infty)$, we obtain,

$$\mathbb{P}(\bar{Y}_n \geq \mu) \leq \max_{x,y\in S} \frac{v_{\theta_\mu}(y)}{v_{\theta_\mu}(x)} \exp\left\{-nD\left(\mu \,\|\, \mu(0)\right)\right\},$$

We assumed that $F$ is closed, and moreover $F$ is bounded since it is a subset of the bounded open interval $\mathcal{M}$. Therefore, $F$ is compact, and so $\dot{\Lambda}^{-1}(F)$ is compact as well. Then due to the fact that $\theta \mapsto v_\theta(x)/v_\theta(y)$ is continuous, from Lemma 2 in [31], we deduce that,

$$\sup_{\theta\in\dot{\Lambda}^{-1}(F)} \max_{x,y\in S} \frac{v_\theta(y)}{v_\theta(x)} < \infty,$$

which we define to be the finite constant $C_+$ of Lemma 2, and which may only depend on $P$, $f$ and $F$. ☐

*Remark* 4. This bound is a variant of Theorem 1 in [31], where the authors derive a Chernoff bound under some structural assumptions on the transition probability matrix $P$ and the function $f$. In our Lemma 2, following their techniques, we derive a Chernoff bound without any assumptions, relying though on the fact that $\mu$ lies in a closed subset of the mean parameter space.

Next, we we present an exponential martingale for Markov chains, which in turn leads to a maximal inequality.

**Lemma 3** (Exponential martingale for Markov chains). *Let $\{X_n\}_{n \in \mathbb{Z}_{>0}}$ be a Markov chain over the finite state space $S$ with an irreducible transition matrix $P$ and initial distribution $q$. Let $f : S \to \mathbb{R}$ be a nonconstant real-valued function on the state space. Fix $\theta \in \mathbb{R}$ and define,*

$$M_n^\theta = \frac{v_\theta(X_n)}{v_\theta(X_0)} \exp\left\{\theta(f(X_1) + \ldots + f(X_n)) - n\Lambda(\theta)\right\}. \tag{10}$$

*Then $\{M_n^\theta\}_{n \in \mathbb{Z}_{>0}}$ is a martingale with respect to the filtration $\{\mathcal{F}_n\}_{n \in \mathbb{Z}_{>0}}$, where $\mathcal{F}_n$ is the $\sigma$-field generated by $X_0, \ldots, X_n$.*

*Proof.*

$$\begin{aligned}
\mathbb{E}(M_{n+1}^\theta \mid \mathcal{F}_n) &= M_n^\theta \frac{e^{-\Lambda(\theta)}}{v_\theta(X_n)} \mathbb{E}(v_\theta(X_{n+1})e^{\theta f(X_{n+1})} \mid \mathcal{F}_n) \\
&= M_n^\theta \frac{e^{-\Lambda(\theta)}}{v_\theta(X_n)} \sum_{x \in S} v_\theta(x) e^{\theta f(x)} P(X_n, y) \\
&= M_n^\theta \frac{e^{-\Lambda(\theta)}}{v_\theta(X_n)} \sum_{x \in S} \tilde{P}_\theta(X_n, x) v_\theta(x) \\
&= M_n^\theta,
\end{aligned}$$

where in the last equality we used the fact that $v_\theta$ is a right Perron-Frobenius eigenvector of $\tilde{P}_\theta$. $\square$

*Proof of Lemma 1.*
Our proof extends the argument from Lemma 11 in [7], which deals with IID random variables. In order to handle the Markovian dependence we need to use the exponential martingale for Markov chains from Lemma 3, as well as continuity results for the right Perron-Frobenius eigenvector.

Following the proof strategy used to establish the law of the iterated logarithm, we split the range of the union $[n]$ into chunks of exponentially increasing sizes. Denote by $\alpha > 1$ the growth factor, to be specified later, and let $n_m = \lfloor \alpha^m \rfloor$ be the end point of the $m$-th chunk, with $n_0 = 0$. An upper bound on the number of chunks is $M = \lceil \log n / \log \alpha \rceil$, and so we have that

$$\bigcup_{k=1}^n \left\{\mu(0) \geq \bar{Y}_k,\ kD\left(\bar{Y}_k \parallel \mu(0)\right) \geq \epsilon\right\} \subseteq \bigcup_{m=1}^M \bigcup_{k=n_{m-1}+1}^{n_m} \left\{\mu(0) \geq \bar{Y}_k, kD\left(\bar{Y}_k \parallel \mu(0)\right) \geq \epsilon\right\}$$

$$\subseteq \bigcup_{m=1}^M \bigcup_{k=n_{m-1}+1}^{n_m} \left\{\mu(0) \geq \bar{Y}_k, D\left(\bar{Y}_k \parallel \mu(0)\right) \geq \frac{\epsilon}{n_m}\right\}.$$

Let $\mu_m = \inf\{\mu < \mu(0) : D\left(\mu \parallel \mu(0)\right) \leq \epsilon/n_m\}$, and $\theta_m = \dot{\Lambda}^{-1}(\mu_m) < \dot{\Lambda}^{-1}(\mu(0)) = 0$ so that $\theta_m \mu_m - \Lambda(\theta_m) = D\left(\mu_m \parallel \mu(0)\right)$. Then,

$$\begin{aligned}
\left\{\mu(0) \geq \bar{Y}_k,\ D\left(\bar{Y}_k \parallel \mu(0)\right) \geq \frac{\epsilon}{n_m}\right\} &\subseteq \left\{\bar{Y}_k \leq \mu_m\right\} \\
&= \left\{e^{\theta_m k \bar{Y}_k - k\Lambda(\theta_m)} \geq e^{k(\theta_m \mu_m - \Lambda(\theta_m))}\right\} \\
&= \left\{M_k^{\theta_m} \geq \frac{v_{\theta_m}(X_k)}{v_{\theta_m}(X_0)} e^{kD(\mu_m \parallel \mu(0))}\right\} \\
&\subseteq \left\{M_k^{\theta_m} \geq \frac{v_{\theta_m}(X_k)}{v_{\theta_m}(X_0)} e^{(n_{m-1}+1)D(\mu_m \parallel \mu(0))}\right\}.
\end{aligned}$$

At this point we use the assumption that $P$ is $(\arg\min_{x\in S} f(x))$-Doeblin in order to invoke Proposition 1 from [31], which in our setting states that there exists a constant $C_- = C_-(P,f) \geq 1$ such that,

$$\frac{1}{C_-} \leq \inf_{\theta\in\mathbb{R}_{\leq 0}, x,y\in S} \frac{v_\theta(y)}{v_\theta(x)}.$$

This gives us the inclusion,

$$\left\{ M_k^{\theta_m} \geq \frac{v_{\theta_m}(X_k)}{v_{\theta_m}(X_0)} e^{(n_{m-1}+1)D(\mu_m \,\|\, \mu(0))} \right\} \subseteq \left\{ M_k^{\theta_m} \geq \frac{e^{(n_{m-1}+1)D(\mu_m \,\|\, \mu(0))}}{C_-} \right\}.$$

In [Lemma 3](#) we have established that $M_k^{\theta_m}$ is a positive martingale, which combined with a maximal inequality for martingales due to [40] (see Exercise 4.8.2 in [11] for a modern reference), yields that,

$$\mathbb{P}\left( \bigcup_{k=n_{m-1}+1}^{n_m} \left\{ M_k^{\theta_m} \geq \frac{e^{(n_{m-1}+1)D(\mu_m \,\|\, \mu(0))}}{C_-} \right\} \right) \leq C_- e^{-(n_{m-1}+1)D(\mu_m \,\|\, \mu(0))}$$

$$\leq C_- e^{-\epsilon \frac{n_{m-1}+1}{n_m}} \leq C_- e^{-\frac{\epsilon}{\alpha}}.$$

To conclude, we pick the growth factor $\alpha = \epsilon/(\epsilon-1)$, and we upper bound the number of chunks by $M \leq \lceil \epsilon\log n \rceil$. $\qquad\square$

## Appendix B   Concentration Properties of Upper Confidence Bounds and Sample Means

**Lemma 4.** *For every arm $a = 1,\ldots,K$, and $t \geq 3$, we have that,*

$$\mathbb{P}_{\theta_a}\left( \min_{n=1,\ldots,t} U_n^a(t) \leq \mu(\theta_a) \right) \leq \frac{4eC_-^a}{t\log t}, \tag{11}$$

*where $C_-^a$ is the constant prescribed in [Lemma 1](#), when the maximal inequality is applied to the Markov chain with parameter $\theta_a$.*

*Proof.*

$$\mathbb{P}_{\theta_a}\left( \min_{n=1,\ldots,t} U_n^a(t) \leq \mu(\theta_a) \right) \leq \mathbb{P}_{\theta_a}\left( \bigcup_{n=1}^{t} \{ \mu(\theta_a) > \bar{Y}_n^a \text{ and } nD\left( \bar{Y}_n^a \,\|\, \mu(\theta_a) \right) \geq g(t) \} \right)$$

$$\leq C_-^a e\lceil g(t)\log t\rceil e^{-g(t)} \leq 4C_-^a e(\log t)^2 e^{-g(t)} = \frac{4eC_-^a}{t\log t},$$

where for the first inequality we used [Equation 7](#) and the definition of $U_n^a(t)$, while for the second inequality we used [Lemma 1](#). $\qquad\square$

**Lemma 5.** *For every arm $a = 1,\ldots,K$, and for $\mu(\lambda) > \mu(\theta_a)$,*

$$\sum_{n=1}^{\infty} \mathbb{P}_{\theta_a}(\mu(\lambda) \leq U_n^a(T)) \leq \frac{g(T)}{D\left( \mu(\theta_a) \,\|\, \mu(\lambda) \right)} + 1 + 8\sigma_{\theta_a,\lambda}^2 \left( \frac{\dot{D}\left( \mu(\theta_a) \,\|\, \mu(\lambda) \right)}{D\left( \mu(\theta_a) \,\|\, \mu(\lambda) \right)} \right)^2 \tag{12}$$

$$+ 2\sqrt{2\pi\sigma_{\theta_a,\lambda}^2} \sqrt{\frac{\dot{D}\left( \mu(\theta_a) \,\|\, \mu(\lambda) \right)^2}{D\left( \mu(\theta_a) \,\|\, \mu(\lambda) \right)^3}} \sqrt{g(T)},$$

*where $\sigma_{\theta,\lambda}^2 = \sup_{\theta\in[\theta_a,\lambda]} \ddot{\Lambda}(\theta) \in (0,\infty)$, and $\dot{D}\left( \mu(\theta_a) \,\|\, \mu(\lambda) \right) = \frac{dD(\mu \,\|\, \mu(\lambda))}{d\mu}\big|_{\mu=\mu(\theta_a)}$.*

*Proof.* The proof is based on the argument given in Appendix A.2 of [7], adapted though for the case of Markov chains. If $\mu(\lambda) \leq U_n^a(T)$, and $\bar{Y}_n^a \leq \mu(\lambda)$, then $D\left( \bar{Y}_n^a \,\|\, \mu(\lambda) \right) \leq g(T)/n$. Let $\mu_x = \inf\{\mu \leq \mu(\lambda) : D\left( \mu \,\|\, \mu(\lambda) \right) \leq x\}$. This in turn implies that $D\left( \bar{Y}_n^a \,\|\, \mu(\lambda) \right) \leq$

$D\left(\mu_{g(T)/n} \parallel \mu(\lambda)\right)$, and using the monotonicity of $\mu \mapsto D\left(\mu \parallel \mu(\lambda)\right)$ for $\mu \leq \mu(\lambda)$, we further have that $\bar{Y}_n^a \geq \mu_{g(T)/n}$. This argument shows that,

$$\mathbb{P}_{\theta_a}\left(\mu(\lambda) \leq U_n^a(T)\right) \leq \mathbb{P}_{\theta_a}\left(\mu_{g(T)/n} \leq \bar{Y}_n^a\right).$$

Therefore,

$$\sum_{n=1}^{\infty} \mathbb{P}_{\theta_a}\left(\mu(\lambda) \leq U_n^a(T)\right) \leq \frac{g(T)}{D\left(\mu(\theta_a) \parallel \mu(\lambda)\right)} + 1 + \sum_{n=n_0+1}^{\infty} \mathbb{P}_{\theta_a}\left(\mu_{g(T)/n} \leq \bar{Y}_n^a\right),$$

where $n_0 = \left\lceil \frac{g(T)}{D\left(\mu(\theta_a) \parallel \mu(\lambda)\right)} \right\rceil$.

Fix $n \geq n_0 + 1$. Then $D\left(\mu(\theta_a) \parallel \mu(\lambda)\right) > g(T)/n$, and therefore $\mu_{g(T)/n} > \mu(\theta_a)$. Furthermore note that $\mu_{g(T)/n}$ is increasing to $\mu(\lambda)$ as $n$ increases, therefore $\mu_{g(T)/n}$ lives in the closed interval $[\mu(\theta_a), \mu(\lambda)]$, and we can apply Lemma 2 for the Markov chain that corresponds to the parameter $\theta_a$,

$$\mathbb{P}_{\theta_a}\left(\bar{Y}_n^a \geq \mu_{g(T)/n}\right) \leq C_+^a e^{-nD\left(\mu_{g(T)/n} \parallel \mu(\theta_a)\right)}.$$

Thus we are left with the task of controlling the sum,

$$\sum_{n=n_0+1}^{\infty} e^{-nD\left(\mu_{g(T)/n} \parallel \mu(\theta_a)\right)}.$$

First note that by definition $\mu_{g(T)/n}$ is increasing in $n$, therefore $D\left(\mu_{g(T)/n} \parallel \mu(\theta_a)\right)$ is positive and increasing in $n$, hence we can perform the following integral bound,

$$\sum_{n=n_0+1}^{\infty} e^{-nD\left(\mu_{g(T)/n} \parallel \mu(\theta_a)\right)} \leq \int_{\frac{g(T)}{D\left(\mu(\theta_a) \parallel \mu(\lambda)\right)}}^{\infty} e^{-sD\left(\mu_{g(T)/s} \parallel \mu(\theta_a)\right)} ds$$

$$= g(T) \int_0^{D\left(\mu(\theta_a) \parallel \mu(\lambda)\right)} \frac{1}{x^2} e^{-\frac{g(T)}{x}D\left(\mu_x \parallel \mu(\theta_a)\right)} dx. \quad (13)$$

The function $\mu \mapsto D\left(\mu \parallel \mu(\lambda)\right)$ is convex thus,

$$D\left(\mu \parallel \mu(\lambda)\right) \geq D\left(\mu(\theta_a) \parallel \mu(\lambda)\right) + \dot{D}\left(\mu(\theta_a) \parallel \mu(\lambda)\right)(\mu - \mu(\theta_a)),$$

where $\dot{D}\left(\mu(\theta_a) \parallel \mu(\lambda)\right) = \frac{dD\left(\mu \parallel \mu(\lambda)\right)}{d\mu}\big|_{\mu=\mu(\theta_a)}$. Plugging in $\mu = \mu_x \geq \mu(\theta_a)$, for $x \in [0, D\left(\mu(\theta_a) \parallel \mu(\lambda)\right)]$, we obtain

$$D\left(\mu(\theta_a) \parallel \mu(\lambda)\right) - x \leq \dot{D}\left(\mu(\theta_a) \parallel \mu(\lambda)\right)(\mu(\theta_a) - \mu_x). \quad (14)$$

From Lemma 8 in [31] we have that,

$$D\left(\mu_x \parallel \mu(\theta_a)\right) \geq \frac{(\mu_x - \mu(\theta_a))^2}{2\sigma_{\theta_a,\lambda}^2}, \quad (15)$$

where $\sigma_{\theta_a,\lambda}^2 = \sup_{\theta \in [\theta_a,\lambda]} \ddot{\Lambda}(\theta) \in (0,\infty)$.

Combining Equation 14 and Equation 15 we deduce that,

$$D\left(\mu_x \parallel \mu(\theta_a)\right) \geq \left(\frac{D\left(\mu(\theta_a) \parallel \mu(\lambda)\right) - x}{\sqrt{2}\sigma_{\theta_a,\lambda}\dot{D}\left(\mu(\theta_a) \parallel \mu(\lambda)\right)}\right)^2.$$

Now we use this bound and break the integral in Equation 13 in two regions, $I_1 = [0, D\left(\mu(\theta_a) \parallel \mu(\lambda)\right)/2]$ and $I_2 = [D\left(\mu(\theta_a) \parallel \mu(\lambda)\right)/2, D\left(\mu(\theta_a) \parallel \mu(\lambda)\right)]$. In the first region we use the fact that $x \leq D\left(\mu(\theta_a) \parallel \mu(\lambda)\right)/2$ to deduce that,

$$\int_{I_1} \frac{1}{x^2} e^{-\frac{g(T)}{x}D\left(\mu_x \parallel \mu(\theta_a)\right)} dx \leq \int_{I_1} \frac{1}{x^2} \exp\left\{-\frac{g(T)}{8\sigma_{\theta_a,\lambda}^2 x}\left(\frac{D\left(\mu(\theta_a) \parallel \mu(\lambda)\right)}{\dot{D}\left(\mu(\theta_a) \parallel \mu(\lambda)\right)}\right)^2\right\} dx$$

$$\leq \frac{8\sigma_{\theta_a,\lambda}^2}{g(T)}\left(\frac{\dot{D}\left(\mu(\theta_a) \parallel \mu(\lambda)\right)}{D\left(\mu(\theta_a) \parallel \mu(\lambda)\right)}\right)^2.$$

In the second region we use the fact that $D\left(\mu(\theta_a) \parallel \mu(\lambda)\right)/2 \leq x \leq D\left(\mu(\theta_a) \parallel \mu(\lambda)\right)$ to deduce that,

$$
\begin{aligned}
\int_{I_2} \frac{1}{x^2} e^{-\frac{g(T)}{x} D(\mu_x \parallel \mu(\theta_a))} dx &\leq \int_{I_2} \frac{4 \exp\left\{-\frac{(x - D(\mu(\theta_a) \parallel \mu(\lambda)))^2}{2\Sigma_{\theta_a,\lambda}}\right\}}{D\left(\mu(\theta_a) \parallel \mu(\lambda)\right)^2} dx \\
&\leq \int_{-\infty}^{D(\mu(\theta_a) \parallel \mu(\lambda))} \frac{4 \exp\left\{-\frac{(x - D(\mu(\theta_a) \parallel \mu(\lambda)))^2}{2\Sigma_{\theta_a,\lambda}}\right\}}{D\left(\mu(\theta_a) \parallel \mu(\lambda)\right)^2} dx \\
&= \frac{2\sqrt{2\pi\sigma_{\theta_a,\lambda}^2}}{\sqrt{g(T)}} \sqrt{\frac{\dot{D}\left(\mu(\theta_a) \parallel \mu(\lambda)\right)^2}{D\left(\mu(\theta_a) \parallel \mu(\lambda)\right)^3}},
\end{aligned}
$$

where $\Sigma_{\theta_a,\lambda} = \frac{\sigma_{\theta_a,\lambda}^2 \dot{D}(\mu(\theta_a) \parallel \mu(\lambda))^2 D(\mu(\theta_a) \parallel \mu(\lambda))}{g(T)}$. $\qquad\square$

**Lemma 6.** *For every arm $a = 1, \ldots, K$,*

$$
\mathbb{P}_{\theta_a}\left(\max_{n=\lceil \delta t \rceil, \ldots, t} |\bar{Y}_n^a - \mu(\theta_a)| \geq \epsilon\right) \leq \frac{c\eta^{\delta t}}{1 - \eta}, \text{ for } \delta \in (0,1), \ \epsilon > 0, \tag{16}
$$

*where $\eta = \eta(\boldsymbol{\theta}, \epsilon) \in (0,1)$, and $c = c(\boldsymbol{\theta}, \epsilon)$ are constants with respect to $t$.*

*Proof.* Using the same technique as in the proof of Lemma 2, we have that for any $\theta \geq 0$ and any $\eta \leq 0$,

$$
\begin{aligned}
\mathbb{P}_{\theta_a}\left(\max_{n=\lceil \delta t \rceil, \ldots, t} |\bar{Y}_n^a - \mu(\theta_a)| \geq \epsilon\right) &\leq \sum_{n=\lceil \delta t \rceil}^{\infty} \max_{x,y \in S} \frac{v_\theta^a(y)}{v_\theta^a(x)} e^{-n(\theta(\mu(\theta_a)+\epsilon) - \Lambda_a(\theta))} \\
&\quad + \sum_{n=\lceil \delta t \rceil}^{\infty} \max_{x,y \in S} \frac{v_\eta^a(y)}{v_\eta^a(x)} e^{-n(\eta(\mu(\theta_a)-\epsilon) - \Lambda_a(\eta))},
\end{aligned}
$$

where by $\Lambda_a(\theta)$ we denote the log-Perron-Frobenious eigenvalue generated by $P_{\theta_a}$, and similarly by $v_\theta^a$ the corresponding right Perron-Frobenius eigenvector.

By picking $\theta = \theta_\epsilon^a$ large enough, and $\eta = \eta_\epsilon^a$ small enough, we can ensure that $\theta(\mu(\theta_a)+\epsilon) - \Lambda_a(\theta) > 0$, and $\eta(\mu(\theta_a) - \epsilon) - \Lambda_a(\eta) > 0$, and so there are constants $\eta = \eta(\boldsymbol{\theta}, \epsilon) \in (0,1)$ and $c = c(\boldsymbol{\theta}, \epsilon)$, such that for any $a = 1, \ldots, K$,

$$
\mathbb{P}_{\theta_a}\left(\max_{n=\lceil \delta t \rceil, \ldots, t} |\bar{Y}_n^a - \mu(\theta_a)| \geq \epsilon\right) \leq c \sum_{n=\lceil \delta t \rceil}^{\infty} \eta^n \leq \frac{c\eta^{\delta t}}{1 - \eta}.
$$

$\qquad\square$

# Appendix C    Analysis of Algorithm 1

As a proxy for the regret we will use the following quantity which involves directly the number of times each arm $a \in \{1, \ldots, N\}$ hasn't been played, and the number of times each arm $b \in \{L+1, \ldots, K\}$ has been played,

$$
\tilde{R}_{\boldsymbol{\theta}}^{\boldsymbol{\phi}}(T) = \sum_{a=1}^{N} (\mu(\theta_a) - \mu(\theta_M)) \, \mathbb{E}_{\boldsymbol{\theta}}^{\boldsymbol{\phi}}[T - N_a(T)] + \sum_{b=L+1}^{K} (\mu(\theta_M) - \mu(\theta_b)) \, \mathbb{E}_{\boldsymbol{\theta}}^{\boldsymbol{\phi}}[N_b(T)]. \tag{17}
$$

For the IID case $\tilde{R}_{\boldsymbol{\theta}}^{\boldsymbol{\phi}}(T) = R_{\boldsymbol{\theta}}^{\boldsymbol{\phi}}(T)$, and in the more general Markovian case $\tilde{R}_{\boldsymbol{\theta}}^{\boldsymbol{\phi}}(T)$ is just a constant term apart from the expected regret $R_{\boldsymbol{\theta}}^{\boldsymbol{\phi}}(T)$. Note that a feature that makes the case of multiple plays more delicate than the case of single plays, even for IID rewards, is the presence of the first summand in Equation 17. For this we also need to analyze the number of times each of the best $N$ arms hasn't been played.

**Lemma 7.**

$$\left| R_{\boldsymbol{\theta}}^{\boldsymbol{\phi}}(T) - \tilde{R}_{\boldsymbol{\theta}}^{\boldsymbol{\phi}}(T) \right| \leq \sum_{a=1}^{K} R_a \cdot \sum_{x \in S} |f(x)|,$$

where $R_a = \mathbb{E}_{\theta_a} \left[ \inf\{n \geq 1 : X_{n+1}^a = X_1^a\} \right] < \infty$.

We start the analysis by establishing the relation between the expected regret, Equation 1, and its proxy, Equation 17. For this we will need the following lemma.

**Lemma 8** (Lemma 2.1 in [3]). *Let $\{X_n\}_{n \in \mathbb{Z}_{\geq 0}}$ be a Markov chain on a finite state space $S$, with irreducible transition probability matrix $P$, stationary distribution $\pi$, and initial distribution $q$. Let $\mathcal{F}_n$ be the $\sigma$-field generated by $X_0, \ldots, X_n$. Let $\tau$ be a stopping time with respect to the filtration $\{\mathcal{F}_n\}_{n \in \mathbb{Z}_{\geq 0}}$ such that $\mathbb{E}[\tau] < \infty$. Define $N(x, n)$ to be the number of visits to state $x$ from time $1$ to time $n$, i.e. $N(x, n) = \sum_{k=1}^{n} I\{X_k = x\}$. Then*

$$|\mathbb{E}[N(x, \tau)] - \pi(x)\,\mathbb{E}[\tau]| \leq R, \text{ for } x \in S,$$

where $R = \mathbb{E}[\inf\{n \geq 1 : X_{n+1} = X_1\}] < \infty$.

*Proof of Lemma 7.*
First note that,

$$S_T = \sum_{a=1}^{K} \sum_{x \in S} f(x) N_a(x, N_a(T)).$$

For each $a \in [K]$, using first the triangle inequality, and then Lemma 8 for the stopping time $N_a(T)$, we obtain,

$$\left| \sum_{x \in S} f(x)(\mathbb{E}_{\boldsymbol{\theta}}^{\boldsymbol{\phi}}[N_a(x, N_a(T))] - \pi_{\theta_a}(x)\,\mathbb{E}_{\boldsymbol{\theta}}^{\boldsymbol{\phi}}[N_a(T)]) \right|$$
$$\leq \sum_{x \in S} |f(x)| \left| \mathbb{E}_{\boldsymbol{\theta}}^{\boldsymbol{\phi}}[N_a(x, N_a(T))] - \pi_{\theta_a}(x)\,\mathbb{E}_{\boldsymbol{\theta}}^{\boldsymbol{\phi}}[N_a(T)] \right|$$
$$\leq R_a \cdot \sum_{x \in S} |f(x)|.$$

Hence summing over $a \in [K]$, and using the triangle inequality, we see that,

$$\left| S_T - \sum_{a=1}^{K} \mu(\theta_a)\,\mathbb{E}_{\boldsymbol{\theta}}^{\boldsymbol{\phi}}[N_a(T)] \right| \leq \sum_{a=1}^{K} R_a \cdot \sum_{x \in S} |f(x)|.$$

To conclude the proof note that,

$$T \sum_{a=1}^{M} \mu(\theta_a) - \sum_{a=1}^{K} \mu(\theta_a)\,\mathbb{E}_{\boldsymbol{\theta}}^{\boldsymbol{\phi}}[N_a(T)]$$
$$= \sum_{a=1}^{N} \mu(\theta_a)\,\mathbb{E}_{\boldsymbol{\theta}}^{\boldsymbol{\phi}}[T - N_a(T)] + \mu(\theta_M)(M - N) - \mu(\theta_M) \sum_{a=N+1}^{K} \mathbb{E}_{\boldsymbol{\theta}}^{\boldsymbol{\phi}}[N_a(T)]$$
$$+ \sum_{b=L+1}^{K} (\mu(\theta_M) - \mu(\theta_b))\,\mathbb{E}_{\boldsymbol{\theta}}^{\boldsymbol{\phi}}[N_b(T)]$$
$$= \sum_{a=1}^{N} (\mu(\theta_a) - \mu(\theta_M))\,\mathbb{E}_{\boldsymbol{\theta}}^{\boldsymbol{\phi}}[T - N_a(T)] + \sum_{b=L+1}^{K} (\mu(\theta_M) - \mu(\theta_b))\,\mathbb{E}_{\boldsymbol{\theta}}^{\boldsymbol{\phi}}[N_b(T)],$$

where in the last equality we used the fact that $\sum_{a=1}^{N} \mathbb{E}_{\boldsymbol{\theta}}^{\boldsymbol{\phi}}[N_a(T)] + \sum_{a=N+1}^{K} \mathbb{E}_{\boldsymbol{\theta}}^{\boldsymbol{\phi}}[N_a(T)] = TM$. $\quad\square$

Next we show that Algorithm 1 is well-defined.

**Proposition 1.** *For each $t \geq K$ we have that $|W_t| \geq M$, and so Algorithm 1 is well defined.*

*Proof of Proposition 1.*
Recall that $\sum_{a \in [K]} N_a(t) = tM$, and so there exists an arm $a_1$ such that $N_{a_1}(t) \geq tM/K$. Then $\sum_{a \in [K] - \{a_1\}} N_a(t) \geq t(M-1)$, and so there exists an arm $a_2 \neq a_1$ such that $N_{a_2}(t) \geq t(M-1)/(K-1)$. Inductively we can see that there exist $M$ distinct arms $a_1, \ldots, a_M$ such that $N_{a_i}(t) \geq t(M-i+1)/(K-i+1) \geq t/K > \delta t$, for $i = 1, \ldots, M$. $\square$

### C.1 Sketch for the rest of the analysis

Due to Lemma 7, it suffices to upper bound the proxy for the expected regret given in Equation 17. Therefore, we can break the analysis in two parts: upper bounding $\mathbb{E}_{\boldsymbol{\theta}}^{\boldsymbol{\phi}}[T - N_a(T)]$, for $a = 1, \ldots, N$, and upper bounding $\mathbb{E}_{\boldsymbol{\theta}}^{\boldsymbol{\phi}}[N_b(T)]$, for $b = L+1, \ldots, K$.

For the first part, we show in Appendix C that the expected number of times that an arm $a \in \{1, \ldots, N\}$ hasn't been played, is of the order of $O(\log \log T)$.

**Lemma 9.** *For every arm $a = 1, \ldots, N$,*

$$\mathbb{E}_{\boldsymbol{\theta}}^{\boldsymbol{\phi}}[T - N_a(T)] \leq \frac{4e\gamma^2 NC \left\lceil \frac{2 \log \gamma}{\log \frac{1}{\delta}} \right\rceil}{\log \gamma} \log \log T + \gamma^{r_0} + \frac{c\gamma^2 \eta^{\delta} K}{(1-\eta)(1-\eta^{\delta})^3},$$

*where $\gamma, r_0, \eta, c$ and $C$ are constants with respect to $T$.*

For the second part, if $b \in \{L+1, \ldots, K\}$, and $b \in \phi_{t+1}$, then there are three possibilities:

1. $L_t \subseteq [L]$, and $|\bar{Y}_a(t) - \mu(\theta_a)| \geq \epsilon$ for some $a \in L_t$,
2. $L_t \subseteq [L]$, and $|\bar{Y}_a(t) - \mu(\theta_a)| < \epsilon$ for all $a \in L_t$, and $b \in \phi_{t+1}$,
3. $L_t \cap \{L+1, \ldots, K\} \neq \emptyset$.

This means that,

$$\mathbb{E}_{\boldsymbol{\theta}}^{\boldsymbol{\phi}}[N_b(T)] \leq M + \sum_{t=K}^{T-1} \mathbb{P}_{\boldsymbol{\theta}}^{\boldsymbol{\phi}} \left( L_t \subseteq [L], \text{ and } |\bar{Y}_a(t) - \mu(\theta_a)| \geq \epsilon \text{ for some } a \in L_t \right)$$

$$+ \sum_{t=K}^{T-1} \mathbb{P}_{\boldsymbol{\theta}}^{\boldsymbol{\phi}} \left( L_t \subseteq [L], \text{ and } |\bar{Y}_a(t) - \mu(\theta_a)| < \epsilon \text{ for all } a \in L_t, \text{ and } b \in \phi_{t+1} \right)$$

$$+ \sum_{t=K}^{T-1} \mathbb{P}_{\boldsymbol{\theta}}^{\boldsymbol{\phi}}(L_t \cap \{L+1, \ldots, K\} \neq \emptyset),$$

and we handle each of those three terms separately.

We show that the first term is upper bounded by $O(1)$.

**Lemma 10.**

$$\sum_{t=K}^{T-1} \mathbb{P}_{\boldsymbol{\theta}}^{\boldsymbol{\phi}} \left( L_t \subseteq [L], \text{ and } |\bar{Y}_a(t) - \mu(\theta_a)| \geq \epsilon \text{ for some } a \in L_t \right) \leq \frac{cL\eta^{\delta K}}{(1-\eta)(1-\eta^{\delta})},$$

*where $c$ and $\eta$ are constant with respect to $T$.*

The second term is of the order of $O(\log T)$, and it is the term that causes the overall logarithmic regret.

**Lemma 11.**

$$\sum_{t=K}^{T-1} \mathbb{P}^{\phi}_{\boldsymbol{\theta}} \left(L_t \subseteq [L], \text{ and } |\bar{Y}_a(t) - \mu(\theta_a)| < \epsilon \text{ for all } a \in L_t, \text{ and } b \in \phi_{t+1}\right)$$

$$\leq \frac{\log T + 3 \log \log T}{D\left(\mu(\theta_b) \parallel \mu(\theta_M) - \epsilon\right)} + 1 + 8\sigma^2_{\mu(\theta_a),\mu(\theta_M)-\epsilon} \left(\frac{\dot{D}\left(\mu(\theta_b) \parallel \mu(\theta_M) - \epsilon\right)}{D\left(\mu(\theta_b) \parallel \mu(\theta_M) - \epsilon\right)}\right)^2$$

$$+ 2\sqrt{2\pi\sigma^2_{\mu(\theta_a),\mu(\theta_M)-\epsilon}} \sqrt{\frac{\dot{D}\left(\mu(\theta_b) \parallel \mu(\theta_M) - \epsilon\right)^2}{D\left(\mu(\theta_b) \parallel \mu(\theta_M) - \epsilon\right)^3}} \left(\sqrt{\log T} + \sqrt{3 \log \log T}\right),$$

*where* $\sigma^2_{\mu(\theta_a),\mu(\theta_M)-\epsilon}$, *and* $\dot{D}\left(\mu(\theta_b) \parallel \mu(\theta_M) - \epsilon\right) = \frac{dD(\mu \parallel \mu(\theta_M)-\epsilon)}{d\mu}\big|_{\mu=\mu(\theta_b)}$, *are constants with respect to* $T$.

Finally, we show that the third term is upper bounded by $O(\log \log T)$.

**Lemma 12.**

$$\sum_{t=K}^{T-1} \mathbb{P}^{\phi}_{\boldsymbol{\theta}}(L_t \cap \{L+1,\ldots,K\} \neq \emptyset) \leq \frac{4e\gamma^2 LC \left\lceil \frac{2\log\gamma}{\log\frac{1}{\delta}} \right\rceil}{\log \gamma} \log \log T + \gamma^{r_0} + \frac{c\gamma^2 \eta^{\delta} K}{(1-\eta)(1-\eta^{\delta})^3},$$

*where* $\gamma, r_0, \eta, c$ *and* $C$ *are constants with respect to* $T$.

This concludes the proof of Theorem 1, modulo the four bounds of this subsection which are established in the next subsection.

## C.2 Proofs for the four bounds

For the rest of the analysis we define the following events which describe good behavior of the sample means and the upper confidence bounds. For $\gamma, r \in \mathbb{Z}_{>1}$ let,

$$A_r = \bigcap_{a \in [K]} \bigcap_{\gamma^{r-1} \leq t \leq \gamma^{r+1}} \left\{ \max_{n=\lceil \delta t \rceil,\ldots,t} |\bar{Y}_n^a - \mu(\theta_a)| < \epsilon \right\},$$

$$B_r = \bigcap_{a \in [N]} \bigcap_{\gamma^{r-1} \leq t \leq \gamma^{r+1}} \left\{ \min_{n=1,\ldots,\lceil \delta t \rceil - 1} U_n^a(t) > \mu(\theta_N) \right\},$$

$$C_r = \bigcap_{a \in [L]} \bigcap_{\gamma^{r-1} \leq t \leq \gamma^{r+1}} \left\{ \min_{n=1,\ldots,\lceil \delta t \rceil - 1} U_n^a(t) > \mu(\theta_a) \right\}.$$

Indeed, the following bounds, which rely on the concentration results of Section 3, suggest that those events will happen with some good probability.

**Lemma 13.**

$$\mathbb{P}_{\boldsymbol{\theta}}(A_r^c) \leq \frac{cK\eta^{\delta\gamma^{r-1}}}{(1-\eta)(1-\eta^{\delta})}, \quad \mathbb{P}_{\boldsymbol{\theta}}(B_r^c) \leq \frac{4eNC \left\lceil \frac{2\log\gamma}{\log\frac{1}{\delta}} \right\rceil}{(r-1)\gamma^{r-1}\log\gamma}, \quad \mathbb{P}_{\boldsymbol{\theta}}(C_r^c) \leq \frac{4eLC \left\lceil \frac{2\log\gamma}{\log\frac{1}{\delta}} \right\rceil}{(r-1)\gamma^{r-1}\log\gamma},$$

*where* $\eta \in (0,1)$, $c$ *and* $C$ *are constants with respect to* $r$.

*Proof.* The first bound follows directly from Equation 16 and a union bound.

For the second bound, let $p = \left\lceil \frac{2\log\gamma}{\log\frac{1}{\delta}} \right\rceil$, so that $\left\lfloor \frac{\gamma^{r-1}}{\delta^p} \right\rfloor \geq \gamma^{r+1}$. For $i = 0,\ldots,p$ let $t_i = \left\lfloor \frac{\gamma^{r-1}}{\delta^i} \right\rfloor$, and define,

$$D_i = \bigcap_{a \in [N]} \left\{ \min_{n=1,\ldots,t_i} U_n^a(t) > \mu(\theta_a) \right\}.$$

From Equation 11 we see that,

$$\mathbb{P}_{\boldsymbol{\theta}}(D_i^c) \leq \frac{4eN \max_{a \in [N]} C_-^a}{t_i \log t_i} \leq \frac{4eN \max_{a \in [N]} C_-^a}{(r-1)\gamma^{r-1}\log\gamma},$$

where $C_-^a$ is the constant from Lemma 1.

Fix $a \in [N]$, and $\gamma^{r-1} \le t \le \gamma^{r+1}$. There exists $i \in \{0, \ldots, p-1\}$ such that $t_i \le t \le t_{i+1}$, and so $t_i > \delta t_i - 1 \ge \delta t - 1$, which gives that $t_i \ge \lceil \delta t \rceil - 1$. On $D_i$, due to Equation 8, we have that,

$$\min_{n=1,\ldots,\lceil \delta t \rceil - 1} U_n^a(t) \ge \min_{n=1,\ldots,\lceil \delta t \rceil - 1} U_n^a(t_i) \ge \min_{n=1,\ldots,t_i} U_n^a(t_i) > \mu(\theta_a) \ge \mu(\theta_N).$$

Therefore,

$$\mathbb{P}_{\boldsymbol{\theta}}(B_r^c) \le \sum_{i=0}^{p-1} \mathbb{P}_{\boldsymbol{\theta}}(D_i^c) \le \frac{4eNp \max_{a \in [N]} C_-^a}{(r-1)\gamma^{r-1} \log \gamma}.$$

The third bound is established along the same lines. $\qquad \square$

In order to establish Lemma 9 we need the following lemma which states that, on $A_r \cap B_r$, an event of sufficiently large probability according to Lemma 13, all the best $N$ arms are played.

**Lemma 14** (Lemma 5.3 in [2]). *Fix* $\gamma \ge \lceil (1 - K\delta)^{-1} \rceil + 2$, *and let* $r_0 = \lceil \log_\gamma \frac{2K}{1 - K\delta - \gamma^{-1}} \rceil + 2$. *For any* $r \ge r_0$, *on* $A_r \cap B_r$ *we have that* $[N] \subset \phi_{t+1}$ *for all* $\gamma^r \le t \le \gamma^{r+1}$.

*Proof of Lemma 9.*

$$
\mathbb{E}_{\boldsymbol{\theta}}^{\boldsymbol{\phi}}[T - N_a(T)] \le \gamma^{r_0} + \sum_{r=r_0}^{\lceil \log_\gamma(T-1) \rceil - 1} \sum_{\gamma^r \le t \le \gamma^{r+1}} \mathbb{P}_{\boldsymbol{\theta}}^{\boldsymbol{\phi}}(a \notin \phi_{t+1})
$$

$$
\le \gamma^{r_0} + \sum_{r=r_0}^{\lceil \log_\gamma(T-1) \rceil - 1} \sum_{\gamma^r \le t \le \gamma^{r+1}} (\mathbb{P}_{\boldsymbol{\theta}}(A_r^c) + \mathbb{P}_{\boldsymbol{\theta}}(B_r^c))
$$

$$
\le \gamma^{r_0} + \sum_{r=r_0}^{\lceil \log_\gamma(T-1) \rceil - 1} \left( \frac{cK\gamma^{r+1}\eta^{\delta\gamma^{r-1}}}{(1-\eta)(1-\eta^\delta)} + \frac{4e\gamma^2 NC \left\lceil \frac{2\log\gamma}{\log\frac{1}{\delta}} \right\rceil}{(r-1)\log\gamma} \right),
$$

where the second inequality follows from Lemma 14, and the third from Lemma 13. Now we use a simple logarithmic upper bound on the harmonic number to obtain,

$$
\sum_{r=r_0}^{\lceil \log_\gamma(T-1) \rceil - 1} \frac{1}{r-1} \le \sum_{r=3}^{\lceil \log_\gamma(T-1) \rceil - 1} \frac{1}{r-1} \le \log \log_\gamma T \le \log \log T.
$$

Finally, we can upper bound the other summand by a constant, with respect to $T$, in the following way,

$$
\sum_{r=r_0}^{\lceil \log_\gamma(T-1) \rceil - 1} \gamma^{r-1}\eta^{\delta\gamma^{r-1}} \le \sum_{k=1}^{\infty} k\eta^{\delta k} = \frac{\eta^\delta}{(1-\eta^\delta)^2}.
$$

$\qquad \square$

*Proof of Lemma 10.*
Using Equation 16 it is straightforward to see that

$$
\mathbb{P}_{\boldsymbol{\theta}}^{\boldsymbol{\phi}} \left( L_t \subseteq [L], \text{ and } |\bar{Y}_a(t) - \mu(\theta_a)| \ge \epsilon \text{ for some } a \in L_t \right) \le \frac{cL\eta^{\delta t}}{1 - \eta},
$$

and the conclusion follows by summing the geometric series. $\qquad \square$

*Proof of Lemma 11.*
Assume that $L_t \subseteq [L]$, and $|\bar{Y}_a(t) - \mu(\theta_a)| < \epsilon$ for all $a \in L_t$, and $b \in \phi_{t+1}$. Then it must be the case that $b \equiv t + 1 \pmod{K}$, $b \notin L_t$, and $U_b(t) > \min_{a \in L_t} \bar{Y}_a(t) > \min_{a \in L_t} \mu(\theta_a) - \epsilon \ge \mu(\theta_M) - \epsilon$. This shows that,

$$
\mathbb{P}_{\boldsymbol{\theta}}^{\boldsymbol{\phi}} \left( L_t \subseteq [L], \text{ and } |\bar{Y}_a(t) - \mu(\theta_a)| < \epsilon \text{ for all } a \in L_t, \text{ and } b \in \phi_{t+1} \right)
$$
$$
\le \mathbb{P}_{\boldsymbol{\theta}}^{\boldsymbol{\phi}}(b \in \phi_{t+1}, \text{ and } U_b(t) > \mu(\theta_M) - \epsilon).
$$

Furthermore,

$$
\sum_{t=K}^{T-1} \mathbb{P}_{\boldsymbol{\theta}}^{\boldsymbol{\phi}}(b \in \phi_{t+1}, \text{ and } U_b(t) > \mu(\theta_M) - \epsilon)
$$

$$
= \sum_{t=K}^{T-1} \sum_{n=M+1}^{M+T-K} \mathbb{P}_{\boldsymbol{\theta}}^{\boldsymbol{\phi}}(\tau_n^b = t+1, \text{ and } U_n^b(t) > \mu(\theta_M) - \epsilon)
$$

$$
\leq \sum_{t=K}^{T-1} \sum_{n=M+1}^{M+T-K} \mathbb{P}_{\boldsymbol{\theta}}^{\boldsymbol{\phi}}(\tau_n^b = t+1, \text{ and } U_n^b(T) > \mu(\theta_M) - \epsilon)
$$

$$
= \sum_{n=M+1}^{M+T-K} \sum_{t=K}^{T-1} \mathbb{P}_{\boldsymbol{\theta}}^{\boldsymbol{\phi}}(\tau_n^b = t+1, \text{ and } U_n^b(T) > \mu(\theta_M) - \epsilon)
$$

$$
\leq \sum_{n=M+1}^{M+T-K} \mathbb{P}_{\theta_b}(U_n^b(T) > \mu(\theta_M) - \epsilon),
$$

where in the first inequality we used Equation 8. Now the conclusion follows from Equation 12. $\square$

In order to establish Lemma 12 we need the following lemma which states that, on $A_r \cap C_r$, an event of sufficiently large probability according to Lemma 13, only arms from $\{1, \dots, L\}$ have been played at least $\lceil \delta t \rceil$ times and have a large sample mean.

**Lemma 15** (Lemma 5.3 B in [2]). *Fix* $\gamma \geq \lceil (1 - K\delta)^{-1} \rceil + 2$, *and let* $r_0 = \lceil \log_\gamma \frac{2K}{1 - K\delta - \gamma^{-1}} \rceil + 2$. *For any* $r \geq r_0$, *on* $A_r \cap C_r$ *we have that* $L_t \subseteq [L]$ *for all* $\gamma^r \leq t \leq \gamma^{r+1}$.

*Proof of Lemma 12.*
From Lemma 15 we see that,

$$
\sum_{t=K}^{T-1} \mathbb{P}_{\boldsymbol{\theta}}^{\boldsymbol{\phi}}(L_t \cap \{L+1, \dots, K\} \neq \emptyset) \leq \gamma^{r_0} + \sum_{r=r_0}^{\lceil \log_\gamma(T-1) \rceil - 1} \sum_{\gamma^r \leq t \leq \gamma^{r+1}} (\mathbb{P}_{\boldsymbol{\theta}}(A_r^c) + \mathbb{P}_{\boldsymbol{\theta}}(C_r^c)).
$$

The rest of the calculations are similar with the proof of Lemma 9. $\square$

*Proof of Corollary 1.*
In the finite-time regret bound of Theorem 1 we divide by $\log T$, let $T$ go to $\infty$, and then let $\epsilon$ go to $0$ in order to get,

$$
\limsup_{T \to \infty} \frac{R_{\boldsymbol{\theta}}^{\boldsymbol{\phi}}(T)}{\log T} \leq \sum_{b=L+1}^{K} \frac{\mu(\theta_M) - \mu(\theta_b)}{D(\mu(\theta_b) \parallel \mu(\theta_M))}.
$$

The conclusion now follows by using the asymptotic lower bound from Equation 3. $\square$

*Proof of Theorem 2.*
The proof of Theorem 2 follows along the lines the proof of Theorem 1, by replacing instances of entries of the right Perron-Frobenius eigenvector $v_\theta(x)$ with one, and is thus omitted. $\square$

# Appendix D  General Asymptotic Lower Bound

Recall from Subsection 2.1 the general one-parameter family of Markov chains $\{\mathbb{P}_\theta : \theta \in \Theta\}$, where each Markovian probability law $\mathbb{P}_\theta$ is characterized by an initial distribution $q_\theta$ and a transition probability matrix $P_\theta$. For this family we assume that,

$$
P_\theta \text{ is irreducible for all } \theta \in \Theta. \tag{18}
$$

$$
P_\theta(x, y) > 0 \;\Rightarrow\; P_\lambda(x, y) > 0, \text{ for all } \theta, \lambda \in \Theta, \; x, y \in S. \tag{19}
$$

$$
q_\theta(x) > 0 \;\Rightarrow\; q_\lambda(x), \text{ for all } \theta, \lambda \in \Theta, \; x \in S. \tag{20}
$$

In general it is not necessary that the parameter space $\Theta$ is the whole real line, but it is assumed to satisfy the following denseness condition. For all $\lambda \in \Theta$ and all $\delta > 0$, there exists $\lambda' \in \Theta$ such that,

$$\mu(\lambda) < \mu(\lambda') < \mu(\lambda) + \delta. \tag{21}$$

Furthermore, the Kullback-Leibler divergence rate is assumed to satisfy the following continuity property. For all $\epsilon > 0$, and for all $\theta, \lambda \in \Theta$ such that $\mu(\lambda) > \mu(\theta)$, there exists $\delta > 0$ such that,

$$\mu(\lambda) < \mu(\lambda') < \mu(\lambda) + \delta \ \Rightarrow \ |D\left(\theta \parallel \lambda\right) - D\left(\theta \parallel \lambda'\right)| < \epsilon. \tag{22}$$

An adaptive allocation rule $\boldsymbol{\phi}$ is called *uniformly good* if,

$$R_{\boldsymbol{\theta}}^{\boldsymbol{\phi}}(T) = o(T^{\alpha}), \text{ for all } \boldsymbol{\theta} \in \Theta^K, \text{ and all } \alpha > 0.$$

Under those conditions [3] establish the following asymptotic lower bound.

**Theorem 3** (Theorem 3.1 from [3]). *Assume that the one-parameter family of Markov chains on the finite state space $S$, together with the reward function $f : S \to \mathbb{R}$, satisfy conditions* (18), (19), (20), (21), *and* (22). *Let $\boldsymbol{\phi}$ be a uniformly good allocation rule. Fix a parameter configuration $\boldsymbol{\theta} \in \Theta^K$, and without loss of generality assume that,*

$$\mu(\theta_1) \geq \ldots \geq \mu(\theta_N) > \mu(\theta_{N+1}) \ldots = \mu(\theta_M) = \ldots = \mu(\theta_L) > \mu(\theta_{L+1}) \geq \ldots \geq \mu(\theta_K).$$

*Then for every $b = L+1, \ldots, K$,*

$$\frac{1}{D\left(\theta_b \parallel \theta_M\right)} \leq \liminf_{T \to \infty} \frac{\mathbb{E}_{\boldsymbol{\theta}}^{\boldsymbol{\phi}}\left[N_b(T)\right]}{\log T}.$$

*Consequently,*

$$\sum_{b=L+1}^{K} \frac{\mu(\theta_M) - \mu(\theta_b)}{D\left(\theta_b \parallel \theta_M\right)} \leq \liminf_{T \to \infty} \frac{R_{\boldsymbol{\theta}}^{\boldsymbol{\phi}}(T)}{\log T}.$$

Lower bounds on the expected regret of multi-armed bandit problems are established using a change of measure argument, which relies on the adaptive allocation rule being uniformly good. [20] gave the prototypical change of measure argument, for the case of IID rewards, and [3] extended this technique for the case of Markovian rewards. Here we give an alternative simplified proof using the data processing inequality, an idea introduced in [17, 9] for the IID case.

We first set up some notation. Denote by $\mathcal{F}_T$ the $\sigma$-field generated by the random variables $\phi_1, \ldots, \phi_T, \{X_n^1\}_{n=0}^{N_1(T)}, \ldots, \{X_n^K\}_{n=0}^{N_K(T)}$, and let $\mathbb{P}_{\boldsymbol{\theta}}^{\boldsymbol{\phi}} \mid_{\mathcal{F}_T}$ be the restriction of the probability distribution $\mathbb{P}_{\boldsymbol{\theta}}^{\boldsymbol{\phi}}$ on $\mathcal{F}_T$. For two probability distributions $\mathbb{P}$ and $\mathbb{Q}$ over the same measurable space we define the *Kullback-Leibler divergence* between $\mathbb{P}$ and $\mathbb{Q}$ as

$$D\left(\mathbb{P} \parallel \mathbb{Q}\right) = \begin{cases} \mathbb{E}_{\mathbb{P}}\left[\log \frac{d\mathbb{P}}{d\mathbb{Q}}\right], & \text{if } \mathbb{P} \ll \mathbb{Q}, \\ \infty, & \text{otherwise,} \end{cases}$$

where $\frac{d\mathbb{P}}{d\mathbb{Q}}$ denotes the Radon-Nikodym derivative, when $\mathbb{P}$ is absolutely continuous with respect to $\mathbb{Q}$. Note that we have used the same notation as for the Kullback-Leibler divergence rate between two Markov chains, but it should be clear from the arguments whether we refer to the divergence or the divergence rate. For $p, q \in [0, 1]$, the *binary Kullback-Leibler divergence* is denoted by

$$D_2\left(p \parallel q\right) = p \log \frac{p}{q} + (1-p) \log \frac{1-p}{1-q}.$$

The following lemma, from [29], will be crucial in establishing the lower bound.

**Lemma 16** (Lemma 1 in [29]). *Let $\boldsymbol{\theta}, \boldsymbol{\lambda} \in \Theta^K$ be two parameter configurations. Let $\tau$ be a stopping time with respect to $(\mathcal{F}_t)_{t \in \mathbb{Z}_{>0}}$, with $\mathbb{E}_{\boldsymbol{\theta}}^{\boldsymbol{\phi}}[\tau], \ \mathbb{E}_{\boldsymbol{\lambda}}^{\boldsymbol{\phi}}[\tau] < \infty$. Then*

$$D\left(\mathbb{P}_{\boldsymbol{\theta}}^{\boldsymbol{\phi}} \mid_{\mathcal{F}_{\tau}} \ \middle\| \ \mathbb{P}_{\boldsymbol{\lambda}}^{\boldsymbol{\phi}} \mid_{\mathcal{F}_{\tau}}\right) \leq \sum_{a=1}^{K} D\left(q_{\theta_a} \parallel q_{\lambda_a}\right) + \sum_{a=1}^{K} \left(\mathbb{E}_{\boldsymbol{\theta}}^{\boldsymbol{\phi}}[N_a(\tau)] + R_{\theta_a}\right) D\left(\theta_a \parallel \lambda_a\right),$$

*where $R_{\theta_a} = \mathbb{E}_{\theta_a}\left[\inf\{n \geq 1 : X_{n+1}^a = X_1^a\}\right] < \infty$, the first summand is finite due to* (20), *and the second summand is finite due to* (19).

*Proof of Theorem 3.*
Fix $b \in \{L+1, \ldots, K\}$, and $\epsilon > 0$. Due to Equation 21 and Equation 22, there exists $\lambda \in \Theta$ such that
$$\mu(\theta_M) < \mu(\lambda), \text{ and } |D(\theta_b \parallel \theta_M) - D(\theta_b \parallel \lambda)| < \epsilon.$$
We consider the parameter configuration $\boldsymbol{\lambda} = (\lambda_1, \ldots, \lambda_K)$ given by,
$$\lambda_a = \begin{cases} \theta_a, & \text{if } a \neq b, \\ \lambda, & \text{if } a = b. \end{cases}$$
Using Lemma 16 we obtain,
$$D\left(\mathbb{P}_{\boldsymbol{\theta}}^{\boldsymbol{\phi}} \mid_{\mathcal{F}_T} \Big\| \mathbb{P}_{\boldsymbol{\lambda}}^{\boldsymbol{\phi}} \mid_{\mathcal{F}_T}\right) \leq D(q_{\theta_b} \parallel q_\lambda) + R_{\theta_b} D(\theta_b \parallel \lambda) + \mathbb{E}_{\boldsymbol{\theta}}^{\boldsymbol{\phi}}[N_b(T)] D(\theta_b \parallel \lambda).$$
From the data processing inequality, see the book of [10], we have that for any event $\mathcal{E} \in \mathcal{F}_T$,
$$D_2\left(\mathbb{P}_{\boldsymbol{\theta}}^{\boldsymbol{\phi}}(\mathcal{E}) \Big\| \mathbb{P}_{\boldsymbol{\lambda}}^{\boldsymbol{\phi}}(\mathcal{E})\right) \leq D\left(\mathbb{P}_{\boldsymbol{\theta}}^{\boldsymbol{\phi}} \mid_{\mathcal{F}_T} \Big\| \mathbb{P}_{\boldsymbol{\lambda}}^{\boldsymbol{\phi}} \mid_{\mathcal{F}_T}\right).$$
We select $\mathcal{E} = \{N_b(T) \geq \sqrt{T}\}$. Then using Markov's inequality, and the fact that $\phi$ is uniformly good we obtain for any $\alpha > 0$,
$$\mathbb{P}_{\boldsymbol{\theta}}^{\boldsymbol{\phi}}(\mathcal{E}) \leq \frac{\mathbb{E}_{\boldsymbol{\theta}}^{\boldsymbol{\phi}}[N_b(T)]}{\sqrt{T}} = \frac{o(T^\alpha)}{\sqrt{T}}, \quad \mathbb{P}_{\boldsymbol{\lambda}}^{\boldsymbol{\phi}}(\mathcal{E}^c) \leq \frac{\mathbb{E}_{\boldsymbol{\lambda}}^{\boldsymbol{\phi}}[T - N_b(T)]}{T - \sqrt{T}} = \frac{o(T^\alpha)}{T - \sqrt{T}}.$$
Using those two inequalities we see that,
$$\liminf_{T \to \infty} \frac{D_2\left(\mathbb{P}_{\boldsymbol{\theta}}^{\boldsymbol{\phi}}(\mathcal{E}) \Big\| \mathbb{P}_{\boldsymbol{\lambda}}^{\boldsymbol{\phi}}(\mathcal{E})\right)}{\log T} = \liminf_{T \to \infty} \frac{\log \frac{1}{\mathbb{P}_{\boldsymbol{\lambda}}^{\boldsymbol{\phi}}(\mathcal{E}^c)}}{\log T} \geq \lim_{T \to \infty} \frac{\log \frac{T - \sqrt{T}}{o(T^\alpha)}}{\log T} = 1.$$
Therefore,
$$\liminf_{T \to \infty} \frac{\mathbb{E}_{\boldsymbol{\theta}}^{\boldsymbol{\phi}}[N^b(T)]}{\log T} \geq \frac{1}{D(\theta_b \parallel \lambda)} \geq \frac{1}{D(\theta_b \parallel \theta_M) + \epsilon},$$
and the first part of Theorem 3 follows by letting $\epsilon$ go to 0. The second part follows from Lemma 7, and Equation 17. $\qquad\square$