[Reviews · NeurIPS 2020]

Review 1

Summary and Contributions: Consider a time discrete world, a set of K random arms, and an operator with a budget of M draws at each time step. Each arm (only when chosen by the operator) advances following a Markov chain parametrized by a single unknown real parameter. The goal of the operator is to design an algorithm that would maximize their gain over a finite time horizon. They first design an analyze an algorithm for this Markovian setting, and prove that it achieves the known lower bound for the problem. Secondly, they show that they can recover from the previous algorithm the iid setting as a special case, and that here as well, the procedure is optimal. Along the way, the authors derive some theoretical tools for ergodic Markov chain among which a maximal inequality (Lemma 2). === AFTER REBUTTAL === Many thanks to the authors for their reply. The reviewer is not satisfied with the answer regarding the main weakness of the paper that he raised. The theoretical contribution is still valuable. The score remains unchanged.

Strengths: - The designed algorithm provably achieves the information theoretical lower bound, making it in a sense not improvable. - The framework offers a unified solution for an exponential family of Markov chains and exponential families in the iid setting. - Lemma 2 could find application for other problems. - The proposed round-robin scheme is computationally more efficient than vanilla UCB adaptive rules.

Weaknesses: In the iid case (Section 5), exponential families are often chosen out of mathematical convenience, for their algebraic properties, existence of sufficient statistics or conjugate priors; but they also seem to arise naturally from nature (e.g. the normal distribution appears in standard limit theorems). In the case of exponential families of chains (Section 4), one requires a base kernel from which the family is derived by exponential tilting. Now, considering such families appears as interesting from an analytical point of view, as it can be shown for example that the rate for large deviations is closely related to the log Perron-Frobenius eigenvalue [42]. However, the main question of this reviewer is perhaps the following. Here the mathematical convenience is certainly beneficial as it leads to a tractable analysis, but beyond this, could it represent a real world problem? Alright, the stationary mean of the chain is tuned by the first derivatve of the log PF eigenvalue, but the whole dynamics of the chain are also constrained by this single parameter, and it is not clear to the reviewer that this captures well large classes of "Markovian arm behaviors". Could the authors motivate their choice of framework by giving an actual example where this assumption would reasonably hold (i.e. there is such a *known* base kernel and every arm is governed by a different tilted version of this base kernel)? The reviewer would be happy to raise their score in this case. [42] A. Dembo and O. Zeitouni, Large Deviations Techniques and Applications, 2nd ed. Springer (1998).

Correctness: Although the reviewer did not verify all the proofs carefully, they found no reason to doubt the results.

Clarity: - Overall, the paper is very neatly written and the organization of the paper is clear. - However, the paper might benefit from another spell-check for good measure. L132 "noncostant" L146 "through out" L270 "initializaton"

Relation to Prior Work: - The difference from prior work is clearly stated by the authors. - This reviewer thinks it is also fair to mention the work of [41], as they also contributed substantially to the development of the theory of exponential families of chains that the authors consider for their bandit setup. [41] Hayashi, Masahito, and Shun Watanabe. "Information geometry approach to parameter estimation in Markov chains." 2014 IEEE International Symposium on Information Theory. IEEE, 2014.

Reproducibility: Yes

Additional Feedback: - Minor details: L162: log 0 = - infinity - It tooks some time to parse the definition at L140-143. It would be an improvement if the authors manage to introduce this notation in a more user-friendly fashion.


Review 2

Summary and Contributions: This paper considers the rested (Markovian) bandit setting, where the learner needs to play M distinct arms in each round. The first result is a maximal inequality for a one-parameter exponential family of Markov chains with finite state space. This result facilitates UCB-scores that are used in the algorithm. The algorithm proceeds by playing the M empirically best arms modulo playing a 'challenger' action (UCB) action in a round-robin fashion. The algorithms is shown to be asymptotically optimal with explicit finite-order terms. A similar result is proposed for the setting of iid rewards where the arm distributions are in an one-dimensional exponential family.

Strengths: The most interesting result to me is the maximal inequality for Markov chains, which to my knowledge is novel. The proof is based on Chernoff's method, with additional assumptions to ensure that the Markov chain reaches it's stationary distribution fast enough. The application to the UCB algorithm follows standard techniques but is non-trivial, hence also a valuable contribution. The round-robin computation of UCB scores / challenger actions is computationally interesting, but has appeared in the literature before (I question the importance of this extension below). Overall, I think this work is of relevance to the NeurIPS community. The author's also empirically compare the methods to standard UCB.

Weaknesses: Update: I agree with the authors that a naive round-robin scheme will not always work, but I wish this contribution could be presented in a more modular fashion (i.e. discussion the standard setting and conditions that allow the round-robin rule). ---- My main objection is the emphasis of the round-robin selection of arms. While I agree that this is computationally attractive and the analysis plays out, I think more generally, one can update the UCB-scores/challenger actions every B rounds, which increases the regret to log(T*B). Hence the additional log(B) regret term is going to disappear asymptotically, but will matter in finite time (also the experiment suggest so). I would like the hear the author's opinion on this. To me it seems that the round-robin plays is an orthogonal idea with a vanishing asymptotically effect, but it should be noted that this will worsen the regret in finite time. If the authors agree, perhaps phrasing the main analysis with the standard way of computing all UCB scores each time and separating the extension to round-robin plays could make the paper easier to follow.

Correctness: The proof follow standard techniques and appears correct to me.

Clarity: Overall the paper is well written, but the paper lacks details and intuition in important places. The first point is the \delta parameter in the algorithm, which I couldn't find explained, neither is it easy to extract it's effect on the regret bounds (it is also not specified for the experiments). Related, some explanation of the set W_t in Algorithm 1 is needed. As a reader, I needed to spend a lot of time trying to understand these lines in the algorithm, and this is still not clear to me. Second, it should be clearly stated that the transition matrix and the function f needs to be known, in order to specify the Markov chain.

Relation to Prior Work: It is unclear to me if Lemma 2 is the first maximal inequality for Markov chains. Please clarify. Other than this, related work is discussed appropriately. The authors give a proof of the asymptotic lower bound in the Appendix; but this is a known result (and stated as such).

Reproducibility: No

Additional Feedback: Line 131: noncostant -> nonconstant


Review 3

Summary and Contributions: This paper considers an interesting extension of the multi-armed bandit problem, where Markov processes generate rewards. The main assumption of the classic multi-armed bandit problem is i.i.d rewards, and the assumption is crucial to analyze the regret lower and upper bounds. This paper relaxes the main assumption using the exponential family of Markov processes and shows the tight regret lower and upper bound. === AFTER REBUTTAL === I appreciate the authors' responses. I've read all the response letter and other reviewers' comments and would like to keep the score.

Strengths: This paper considers an interesting and important problem non-i.i.d rewards bandit. Since the classic MAB results rely on the i.i.d assumption, all the results do not hold when the rewards are not i.i.d. This paper studies how to deal with the Markov rewards. This paper provides the tight regret lower bound and designs the regret optimal algorithm using the KL-UCB concept under the exponential family of Markov reward processes. The concentration inequalities for Markov chains are well discussed. When the observations have dependency like the Markov chains, it is much more difficult to derive concentration inequalities than the i.i.d. cases. This paper review recent results and provide a very tight concentration inequality especially for the exponential family of Markov process.

Weaknesses: The exponential family of Markov process is still a strong assumption although it is much better than i.i.d. Since the assumption is very strong, I would like to know more detailed applications which scenarios follow the exponential family of Markov process.

Correctness: I think the theorems are correct although I didn't check all the details

Clarity: This paper is well written.

Relation to Prior Work: The related work section discuss most of important related works.

Reproducibility: Yes

Additional Feedback: I've read all reviews and the response letter. I think this paper is a good theory paper and would like to keep my score "accept"

[Author Response · NeurIPS 2020]

Dear reviewers, first of all we would like to thank you for taking the time to review our paper during those challenging times! Answers to your questions are in place.

**Exponential family of stochastic matrices:** We sketch two examples illustrating that one-parameter exponential families of stochastic matrices generalize all the applications of exponential families of discrete probability distributions.

1. Take example 1 from the paper. As discussed in the paper this generalizes the Bernoulli exponential family. In the i.i.d. bandit model one would have $K$ Bernoulli processes, $\mathrm{Ber}(p_{\theta_1}), \ldots, \mathrm{Ber}(p_{\theta_K})$. In the Markovian bandit model under consideration one has $K$ Markov processes specified by $K$ stochastic matrices $P_{\theta_1}, \ldots, P_{\theta_K}$, and each row of them is specified by a coin flip as discussed in example 1. Those $K$ Markovian processes form a generalization of the $K$ Bernoulli processes.

2. In the same spirit as before, we will sketch an approximation for the generalization of the Poisson exponential family, which is useful for count data. Due to the finite state space, approximate the $\mathrm{Pois}(\lambda)$ distribution, with $\mathrm{Bin}(n, \lambda/n)$, and $n$ large enough so that the two distributions are $\epsilon$-close with respect to the total variation distance. Now our state space is $S = \{0, \ldots, n\}$. For the generator stochastic matrix we pick $n + 1$ row distributions $\mathrm{Bin}(n, \lambda_0/n), \ldots, \mathrm{Bin}(n, \lambda_n/n)$ which are approximately $\mathrm{Pois}(\lambda_0), \ldots, \mathrm{Pois}(\lambda_n)$, and for the Markovian bandit model we tilt the generator stochastic matrix by parameters $\theta_1, \ldots, \theta_K$, to obtain $K$ Markovian processes, each of them giving rewards and transitioning according to approximate Poisson distributions. In the i.i.d. case we would have just a single $\mathrm{Pois}(\lambda)$ distribution, and we would produce $K$ tilts which would correspond to the distributions of the $K$ arms.

The problem with generalizing even further to countably infinite state spaces, or even continuous state spaces is the peculiar behavior of eigenvalues and eigenfunctions (which may not even exist) on infinite dimensional spaces.

**Round-robin scheme:** As discussed in the paper the round-robin idea dates back to Lai and Robbins, although forgotten nowadays. In this paper:

1. We use different statistics to make the scheme computable in the case of multiple-plays and Markovian rewards (note that coming up with a computable algorithm in the case of multiple-plays is the motivation of [18] Komiyama, Honda, Nakagawa, where they study Thompson sampling for multiple plays and i.i.d. Bernoulli rewards).

2. We provide a finite-time analysis (as opposed to asymptotics in prior work).

3. Through experimental results we bring to the attention of the research community that this type of scheme is computationally more efficient, and equally as effective as the status quo of calculating UCB or KL-UCB indices for each arm at each round. In particular for KL-UCB type of indices the computational improvement can be quantified as $O(K + \log 1/\epsilon)$ (this paper) vs $O(K \log 1/\epsilon)$ (KL-UCB paper) cost per round, where $K$ is the number of arms, and $\epsilon$ is the quality of the KL divergence inverses that we're interested in.

Reviewer #3 we don't think that your suggestion as an alternative to the round-robin scheme works. For instance take $B = T/2$, where $T$ is the time horizon. Then you're claiming that by calculating UCB scores just twice, the regret might only get double, which is clearly wrong. Additionally, the generator stochastic matrix $P$ and the function $f$ need not be know for the algorithm. All need to be known is what declared as parameters in the preamble of the algorithm, so in particular the only thing need to be known from the family is the KL divergence rate function. Finally, we could even eliminate the presence of $\delta$, but we decided to keep it as knob to tune the algorithm. In particular for the experiments we tuned $\delta$ by playing around with several values in the range $(0, 1/K)$.

**Maximal inequality:** The workhorse behind a maximal inequality is typically some variant of Doob's martingale inequality. For our paper we have reviewed the literature on martingale methods to derive deviation inequalities for Markov chains or more general dependent processes [8, 15, 19, 24, 25, 26, 27, 30, 33, 34], and to the best of our knowledge none of them seems capable to deliver a maximal inequality, due to the fact that they don't use the exponential martingale (lemma 1), but instead they use Doob's or Dynkin's martingale. For your interest, A. Kontorovich is one of the authors of [15] and also an AC, so maybe you could consult him for an extra opinion?

[Meta-Review · NeurIPS 2020]

This paper makes a good theoretical progress in the Markovian bandit. Still, the rebuttal to the raised concerns (the model is artificial, the use of round-robin manner is not novel and has cost in finite-time, etc.) is not convincing although the theoretical contribution overtakes them. Thus we expect that the true contribution is presented clearer in the coming version.